# scAAVengr, a transcriptome-based pipeline for quantitative ranking of engineered AAVs with single-cell resolution

Bilge E Öztürk[1], Molly E Johnson[1], Michael Kleyman[2], Serhan Turunç[1], Jing He[3], Sara Jabalameli[1], Zhouhuan Xi[1,4], Meike Visel[5], Valérie L Dufour[6], Simone Iwabe[6], Luis Felipe L Pompeo Marinho[6], Gustavo D Aguirre[6], José-Alain Sahel[1], David V Schaffer[5,7], Andreas R Pfenning[2], John G Flannery[5,8], William A Beltran[6], William R Stauffer[3], Leah C Byrne[1,3,6,9]*

[1]Department of Ophthalmology, University of Pittsburgh, Pittsburgh, United States; [2]Computational Biology, School of Computer Science, Carnegie Mellon University, Pittsburgh, United States; [3]Department of Neurobiology, University of Pittsburgh, Pittsburgh, United States; [4]Eye Center of Xiangya Hospital, Hunan Key Laboratory of Ophthalmology, Central South University, Changsha, China; [5]Helen Wills Neuroscience Institute, University of California, Berkeley, Berkeley, United States; [6]Division of Experimental Retinal Therapies, Department of Clinical Sciences & Advanced Medicine, School of Veterinary Medicine, University of Pennsylvania, Philadelphia, United States; [7]Chemical Engineering, University of California, Berkeley, Berkeley, United States; [8]Vision Science, Herbert Wertheim School of Optometry, University of California Berkeley, Berkeley, United States; [9]Department of Bioengineering, University of Pittsburgh, Pittsburgh, United States

*For correspondence: lbyrne@pitt.edu

## Abstract

**Background:** Adeno-associated virus (AAV)-mediated gene therapies are rapidly advancing to the clinic, and AAV engineering has resulted in vectors with increased ability to deliver therapeutic genes. Although the choice of vector is critical, quantitative comparison of AAVs, especially in large animals, remains challenging.

**Methods:** Here, we developed an efficient single-cell AAV engineering pipeline (scAAVengr) to simultaneously quantify and rank efficiency of competing AAV vectors across all cell types in the same animal.

**Results:** To demonstrate proof-of-concept for the scAAVengr workflow, we quantified – with cell-type resolution – the abilities of naturally occurring and newly engineered AAVs to mediate gene expression in primate retina following intravitreal injection. A top performing variant identified using this pipeline, K912, was used to deliver SaCas9 and edit the rhodopsin gene in macaque retina, resulting in editing efficiency similar to infection rates detected by the scAAVengr workflow. scAAVengr was then used to identify top-performing AAV variants in mouse brain, heart, and liver following systemic injection.

**Conclusions:** These results validate scAAVengr as a powerful method for development of AAV vectors.

**Funding:** This work was supported by funding from the Ford Foundation, NEI/NIH, Research to Prevent Blindness, Foundation Fighting Blindness, UPMC Immune Transplant and Therapy Center, and the Van Sloun fund for canine genetic research.

**eLife digest** Gene therapy is an experimental approach to treating disease that involves altering faulty genes or replacing them with new, working copies. Most often, the new genetic material is delivered into cells using a modified virus that no longer causes disease, called a viral vector. Virus-mediated gene therapies are currently being explored for degenerative eye diseases, such as retinitis pigmentosa, and neurological disorders, like Alzheimer's and Parkinson's disease. A number of gene therapies have also been approved for treating some rare cancers, blood disorders and a childhood form of motor neuron disease.

Despite the promise of virus-mediated gene therapy, there are significant hurdles to its widespread success. Viral vectors need to deliver enough genetic material to the right cells without triggering an immune response or causing serious side effects. Selecting an optimal vector is key to achieving this. A type of viruses called adeno-associated viruses (AAV) are prime candidates, partly because they can be easily engineered. However, accurately comparing the safety and efficacy of newly engineered AAVs is difficult, due to variation between test subjects and the labor and cost involved in careful testing.

Öztürk et al. addressed this issue by developing an experimental pipeline called scAAVengr for comparing gene therapy vectors head-to-head. The process involves tagging potential AAV vectors with unique genetic barcodes, which can then be detected and quantified in individual cells using a technique called single-cell RNA sequencing. This means that when several vectors are used to infect lab-grown cells or a test animal at the same time, they can be tracked. The vectors can then be ranked on their ability to infect specific cell types and deliver useful genetic material.

Using scAAVengr, Öztürk et al. compared viral vectors designed to target the light-sensitive cells of the retina, which allow animals to see. First, a set of promising viral vectors were evaluated using the scAAVengr pipeline in the eyes of marmosets and macaques, two small primates. Precise levels and locations of gene delivery were quantified. The top-performing vector was then identified and used to deliver Cas9, a genome editing tool, to primate retinas.

Öztürk et al. also used scAAVengr to compare viral vectors in mice, analysing the vectors' ability to deliver their genetic cargo to the brain, heart, and liver. These experiments demonstrated that scAAVengr can be used to evaluate vectors in multiple tissues and in different organisms.

In summary, this work outlines a method for identifying and precisely quantifying the performance of top-performing viral vectors for gene therapy. By aiding the selection of optimal viral vectors, the scAAVengr pipeline could help to improve the success of preclinical studies and early clinical trials testing gene therapies.

## Introduction

Gene therapy is a rapidly developing approach for the treatment of inherited disease, and AAV is a leading viral vector candidate for safe and efficient delivery. A growing number of clinical trials are using AAV to treat diseases such as retinal degeneration, neurological disorders, and hemophilia, through gene replacement, genome editing, and optogenetics (*High and Roncarolo, 2019*; *Sahel and Dalkara, 2019*). And, with the recent FDA approval for treatment of Leber congenital amaurosis and spinal muscular atrophy, gene therapies are rapidly becoming a clinical reality. However, significant hurdles prevent the successful, widespread implementation of AAV-mediated gene therapies, including efficient gene delivery and immune response to viral vectors and gene products. Recent efforts to reengineer viral vectors have shown promise for addressing these issues, resulting in AAVs with improved abilities (*Li and Samulski, 2020*). The enhanced viral vectors produced by these high-throughput methods still require quantitative validation and comparison, however, currently a challenging and burdensome process.

The selection of an optimal vector is essential to the success of the therapy. Sufficient gene expression is critical, while enhanced tropism and greater efficiency of gene delivery reduces the titer of vector required and decreases the likelihood of immune response. Quantitative comparisons of newly engineered vectors, including evaluation of transgene expression levels and cell-type tropism, have in the past required large numbers of animals, and therefore involved significant ethical and financial burden. Additionally, in primates, the large variability between animals, due to differences in

anatomy and immune responses, has made comparisons between animals inaccurate. Here, we have developed a single cell RNA-seq AAV engineering (scAAVengr) pipeline for rapid, quantitative in vivo comparison of transgene expression from newly engineered AAV capsid variants across all different cell types in a tissue in parallel, and in the same animals.

In this work, the scAAVengr pipeline was applied to primates, which are the most physiologically similar animal to humans, and are thus a critical preclinical model. Successful clinical translation of gene therapies depends on highly efficient vectors for human tissue, and vector performance in small animals often does not accurately predict efficiency in primates. For retinal gene therapy in particular, primates are essential, as existing AAV vectors infect the primate retina significantly less efficiently than in rodent retina (*Dalkara et al., 2013*). Furthermore, primates are the only animal model that has a macula and foveal pit (the region of the retina responsible for high acuity vision in humans), making them the most relevant translational model. Notably, the pattern of AAV expression also differs in foveal and in peripheral retina (*Dalkara et al., 2013*), with highest expression in the foveola and in a perifoveal ring of retinal ganglion cells, and punctate expression near blood vessels in the periphery. The scAAVengr single-cell RNA-Seq pipeline allowed us to quantitatively evaluate the clinical potential of multiple lead candidates across all retinal cell types, in the foveal and peripheral retina, in a large animal model with eyes similar to humans. The scAAVengr pipeline can be applied to any species or tissue for which marker genes can be identified, however, as demonstrated here through screening performed in mouse brain, heart, and liver following systemic injections of pooled AAV library.

# Materials and methods

## Key resources table

| Reagent type (species) or resource | Designation | Source or reference | Identifiers | Additional information |
|---|---|---|---|---|
| Strain, strain background (*Escherichia coli*) | NEB 5-alpha | NEB | C2987H | Competent cells |
| Strain, strain background (*Escherichia coli*) | MegaX DH10B T1 | ThermoFisher | C640003 | Electrocompetent cells |
| Strain, strain background (*Mus musculus*) | C57Bl/6 J | Jackson Laboratories | Stock No: 000664 RRID:IMSR_JAX:000664 | |
| Cell line (*Homo-sapiens*) | 293AAV | Cell Biolabs | AAV-100 RRID:CVCL_KA64 | |
| Cell line (*Homo-sapiens*) | HEK293T | ATCC | CRL-1573 RRID:CVCL_0045 | |
| Antibody | Lectin PNA (Peanut agglutinin) | Molecular Probes | L32459 RRID:AB_2315178 | (1:200) |
| Antibody | Anti-GFP, (rabbit polyclonal) | Thermofisher Scientific | A11122 RRID:AB_221569 | (1:250) |
| Recombinant DNA reagent | pX601-AAV-CMV::NLS-SaCas9-NLS-3xHA-bGHpA;U6::Bsal-sgRNA | Addgene | Plasmid #61591 RRID:Addgene_61591 | A single vector AAV-Cas9 system containing SaCas9 and its sgRNA |
| Recombinant DNA reagent | scCAG-eGFP-Barcode-bghPolyA | This paper | | Byrne Lab, see materials and methods, under GFP barcoded AAV library construction. |
| Recombinant DNA reagent | AAV libraries | References: ~ 588 peptide insertion library (*Müller et al., 2003*), AAV2-Loopswap library (*Koerber et al., 2008*) AAV2-ErrorProne library (*Koerber et al., 2006*) SCHEMA library (*Ojala et al., 2018*). | | |

*Continued on next page*

*Continued*

| Reagent type (species) or resource | Designation | Source or reference | Identifiers | Additional information |
|---|---|---|---|---|
| Commercial assay or kit | QuickTiter AAV Quantitation Kit | Cell biolabs | VPK-145 | AAV quantification kit |
| Commercial assay or kit | Qiagen DNeasy Blood and Tissue Kit | Qiagen | Cat. No. / ID: 69504 | DNA extraction kit |
| Commercial assay or kit | AllPrep DNA/RNA Micro Kit | Qiagen | Cat. No. / ID: 80284 | DNA/RNA extraction kit |
| Commercial assay or kit | Neural Tissue Dissociation Kit for postnatal neurons | MACS Miltenyi | 130-094-802 | Retina dissociation kit |
| Commercial assay or kit | Adult Brain Tissue Dissociation Kit | MACS Miltenyi | 130-107-677 | Brain dissociation kit |
| Commercial assay or kit | Multi Tissue Dissociation Kit 2 | MACS Miltenyi | 130-110-203 | Heart dissociation kit |
| Commercial assay or kit | Liver Dissociation Kit | MACS Miltenyi | 130-105-807 | Liver dissociation kit |
| Commercial assay or kit | Chromium Next GEM Single Cell 3' Reagent Kits v3 | 10x Genomics | PN-1000075, PN-1000073, PN-120262 | |
| Commercial assay or kit | Chromium Next GEM Single Cell 3' Reagent Kits v3.1 (Dual Index) | 10x Genomics | PN-1000268, PN-1000120, PN-1000215 | |
| Commercial assay or kit | Targeted Gene Expression Reagent Kit | 10x Genomics | PN-1000248, PN-1000249 | |
| Chemical compound, drug | Cyclosporine | GENGRAF | | 6 mg/kg |
| Chemical compound, drug | Meloxicam | Vivlodex | | 0.2 mg/kg |
| Chemical compound, drug | Triamcinolone Acetonide (Kenalog 40) | Bristol-Myers Squibb | | |
| Software, algorithm | STARsolo | A.Dobin et al., STAR: ultrafast universal RNA-seq aligner. Bioinformatics 29, 15–21 (2013). | RRID:SCR_021542 | v2.7 |
| Software, algorithm | Cell Ranger | 10x Genomics | RRID:SCR_017344 | v3 |
| Software, algorithm | DropletUtils | A.T. L. Lun et al., EmptyDrops: distinguishing cells from empty droplets in droplet-based single-cell RNA sequencing data. Genome Biol 20, 63 (2019). | | v1.4.3 |
| Software, algorithm | SoupX | M.D. Young, S. Behjati, SoupX removes ambient RNA contamination from droplet based single cell RNA sequencing data. bioRxiv, (2020). | RRID:SCR_019193 | v0.3.1 |
| Software, algorithm | SCDS | A.S. Bais, D. Kostka, scds: computational annotation of doublets in single-cell RNA sequencing data. Bioinformatics 36, 1150–1158 (2020). | RRID:SCR_021541 | v1.0.0 |
| Software, algorithm | Scran | Lun ATL, McCarthy DJ, Marioni JC. A step-by-step workflow for low-level analysis of single-cell RNA-seq data with Bioconductor. F1000Research 5: 2122 (2016). | RRID:SCR_016944 | v1.12.1 |

*Continued*

| Reagent type (species) or resource | Designation | Source or reference | Identifiers | Additional information |
|---|---|---|---|---|
| Software, algorithm | ALRA | G.C. Linderman, J. Zhao, Y. Kluger, Zero-preserving imputation of scRNA-seq data using low-rank approximation. bioRxiv, (2018). | RRID:SCR_021540 | v1.0 |
| Software, algorithm | Scanpy | F.A. Wolf, P. Angerer, F. J. Theis, SCANPY: large-scale single-cell gene expression data analysis. Genome Biol 19, 15 (2018). | RRID:SCR_018139 | v1.4.4.post1 |
| Software, algorithm | Scanorama | Hie B, Bryson B, Berger B. Efficient integration of heterogeneous single-cell transcriptomes using Scanorama. Nature Biotechnology 37: 685–691 (2019). | RRID:SCR_021539 | v1.2 |
| Software, algorithm | Salmon | R.Patro, G. Duggal, M. I. Love, R. A. Irizarry, C. Kingsford, Salmon provides fast and bias-aware quantification of transcript expression. Nat Methods 14, 417–419 (2017). | RRID:SCR_017036 | v0.9.1 |
| Software, algorithm | CRISPResso2 | Clement K, Rees H, Canver MC, Gehrke JM, Farouni R, Hsu JY, Cole MA, Liu DR, Joung JK, Bauer DE, Pinello L. CRISPResso2 provides accurate and rapid genome editing sequence analysis. Nature Biotechnology 37: 224–226 (2019). | RRID:SCR_021538 | v2.0.34 |

## Study approval

All procedures were performed in compliance with the ARVO statement for the Use of Animals in Ophthalmic and Vision Research, and for canine studies with approval by the University of Pennsylvania Institutional Animal Care and Use Committee (IACUC # 803813), and for the NHP and mouse studies with approval from the University of Pittsburgh Institutional Animal Care and Use Committee (IACUC #18042326).

## Animals

### Dogs

Dogs, between the age of 7 and 17 months, were screened for neutralizing antibodies to AAV2 as previously described (*Day et al., 2018*). All dogs had titers < 1:25. A subconjunctival injection of 4 mg of Triamcinolone Acetonide (Kenalog 40) was performed immediately after intravitreal AAV delivery. Animals were treated post-injection with daily topical application of prednisolone acetate and oral antibiotics and a tapering dose of corticosteroids. Non-invasive retinal imaging by confocal scanning laser ophthalmoscopy (Spectralis HRA + OCT, Heidelberg Engineering, Germany) was performed under general anesthesia. Overlapping en face images were captured using the short-wavelength (480 nm) autofluorescence imaging mode to detect GFP expression. Following euthanasia, retinal samples were collected either for DNA/RNA analysis (see above) or for immunohistochemistry after paraformaldehyde fixation and embedding. The dogs were maintained at the Retinal Disease Studies Facility, Kennett Square, Pennsylvania. Other than uveitis in two eyes, all other injected eyes showed no adverse events (*Supplementary file 1*).

### Primates

Marmosets, cynomolgus macaques and rhesus macaques were between 3–10 years of age, and intra-vitreal injections were made with methods described previously (*Dalkara et al., 2013*). All NHPs used in these studies were previously screened to have anti-AAV2 neutralizing antibody titers of <1:5. Monkeys used for K912-scCAG-GFP fluorophore expression received daily oral doses of cyclosporine at a dose of 6 mg/kg for the duration of the study. Marmosets received oral daily doses of meloxicam (0.2 mg/kg) for 1 week after injection. At the conclusion of the experiment, euthanasia was done with an IV overdose of sodium pentobarbital (75 mg kg−1), as recommended by the Panel on Euthanasia of the American Veterinary Medical Association. A summary of minor adverse events related to the procedures is summarized in *Supplementary file 2*. Other than uveitis in two eyes, all other injected eyes showed no adverse events.

## Mice

C57BL/6 J mice from Jackson Laboratories were used for mouse experiments. 50 µL of pooled AAV vector library was delivered by systemic injections via facial vein injection to P0 mice, which were anesthetized on ice. Tissues were collected 3 weeks after injection. No adverse events were noted.

## AAV packaging

AAV vectors were produced in HEK293T cells (ATCC), or 293AAV cells (Cell Biolabs) using a double (for AAV2-7mer, LoopSwap, AAV2-ErrorProne and SCHEMA libraries) or triple transfection method (*Grieger et al., 2006*). Short tandem repeat profiling was done by ATTC Cell Line Authentication Service and all cell lines were checked for mycoplasma using Hoechst staining. Directed evolution libraries were packaged using an empirically determined molar ratio of plasmids in the packaging cell line, such that each AAV particle contained the genome encoding its own capsid (*Koerber et al., 2006*). All recombinant AAVs were purified by iodixanol gradient ultracentrifugation, buffer exchanged and concentrated with Amicon Ultra-15 Centrifugal Filter Units (#UFC8100) in DPBS and titered by quantitative PCR relative to a standard curve using ITR-binding primers or by using Quick-Titer AAV Quantitation Kit (Cell Biolabs). The relative titer of each variant was confirmed by Illumina MiSeq sequencing.

## Directed evolution performed in canine retina

We packaged AAV2 error prone (*Koerber et al., 2006*), AAV2-7mer (*Müller et al., 2003*), loop swap (*Koerber et al., 2008*) and SCHEMA libraries (*Ojala et al., 2018*), which were pooled and injected intravitreally into both eyes of wild-type dogs (*Figure 1—figure supplement 1*). Intravitreal injections (150–250 µL) were performed with a 30-gauge insulin syringe under general anesthesia delivering the viral solution in the mid-vitreous. Three weeks later, dogs were euthanized by intravenous injection of sodium pentobarbital, and both eyes were flattened by making relief cuts in the globe. Two mm punches of RPE were immediately collected from superior, inferior, temporal, and nasal regions of the retina, as well as from the area centralis, and flash frozen. DNA was extracted from samples using a Qiagen DNeasy blood and tissue kit, according to the manufacturer's instructions, and AAV *cap* genes were recovered via PCR from retinal pigment epithelium (RPE) punches. AAV genomes were then repackaged and reinjected. Five rounds of selection were performed (*Supplementary file 1*), with error prone PCR done following the third round of selection to introduce additional diversity into the library.

## Deep sequencing of directed evolution libraries from rounds of selection conducted in dogs

Following rounds of selection, the AAV2-7mer library was found to give rise to the majority of resulting variants. Each round of selection from the AAV2-7mer was then subjected to deep sequencing in order to analyze the dynamics of each individual variant and overall convergence of the library. A ~ 75–85 base pair region containing the 7mer insertion was PCR amplified from harvested DNA. Primers included Illumina adapter sequences containing unique barcodes to allow for multiplexing of amplicons from multiple rounds of selection (*Supplementary file 4*). PCR amplicons were purified and sequenced with a 100-cycle single-read run on an Illumina HiSeq 2,500. DNA sequences were translated into amino acid sequences, and the number of reads containing unique 7mer insert sequences were counted. Read counts were normalized by the total number of reads in the run. Pandas was used to create plots.

## Deep sequencing analysis of rounds of selection in canine

Best performing variants were chosen as variants with the greatest fold increase in the final round of selection relative to the initial plasmid library (# reads in final round, normalized to total number of reads in the round / # of reads in plasmid library, normalized to total number of reads in the round). A pseudo-count of 1 was added to each variant in every round, in order to mitigate effects of small number increases and allow analysis of variants with a zero count in sequencing of the original library (*Fowler et al., 2014*).

## Construction and analysis of the directed evolution subset library

Twenty top variants with the largest fold increases during the overall selection were chosen for a head-to-head analysis in canine retina (the directed evolution subset library). To compare the selected

variants head-to-head, these 20 vectors, along with an AAV2 control, were packaged individually with a ubiquitous CAG promoter driving expression of GFP fused to a unique DNA barcode (AAV-barcode). Vectors were titer matched, mixed together, and injected intravitreally into both eyes of 3 WT dogs. Six weeks after injection, potent GFP fluorescence was detected by fundus imaging of the canine eyes. GFP expression was present in every layer of the dog retina (*Figure 1—figure supplements 1 and 3*). Eyes were harvested, tissue samples were collected from across the retina, the RPE was separated from the neuroretina, and photoreceptors were collected using transverse sectioning on a cryostat (*Byrne et al., 2020*). DNA and mRNA were extracted from retinal samples using a Qiagen Allprep kit. Samples were collected from areas across the retina, and from the outer nuclear layer (ONL) or RPE. Following DNA and mRNA extraction, AAV-barcodes were PCR amplified from genomic DNA and from cDNA, from photoreceptors and RPE. cDNA was created from mRNA using Superscript III reverse transcriptase, according to the manufacturer's recommendations. AAV-barcodes were PCR amplified directly from DNA or cDNA. Primers amplified a ~ 50 bp region surrounding the AAV-barcode and contained Illumina adapter sequences and secondary barcodes to allow for multiplexing of multiple samples (*Supplementary file 4*).

AAV-barcodes amplified from the ONL, RPE, and the injected AAV libraries were then subjected to Illumina sequencing to quantify the representation of each of the variants. PCR amplicons sequenced with a 100-cycle single-read run on a MiSeq. Read counts were normalized by total number of reads in the run. Analysis of barcode abundance was performed using in-house code written in Python, followed by creation of plots in Pandas. Best performing variants were selected based on the fold increase in the percent of total library, relative to the injected library (% of total in recovered sample / % of total in injected library). Analysis was performed on n = 3 dogs. Variants were ranked on the basis of the normalized change in frequency of their representation in the recovered genomes relative to the injected AAV library (% of total in recovered AAV library / % of total in injected library). Selected variants largely outperformed AAV2 in three dogs and across peripheral, mid-peripheral and central retina (*Figure 1—figure supplement 4*). In addition, the most abundant variant (K91, LAHQDTTKNA), which was overly represented in the original library, did not outperform other variants in canine retina, indicating that the metric of quantity of representation in the final round of selection is not the best indicator of fitness for transgene expression. Rankings based on mRNA and DNA recovery indicated different top-performing variants. Evaluation on the basis of mRNA is a more relevant readout of AAV performance, as it is indicative of transgene expression, rather than persistence in extracellular spaces of the tissue or viral endocytosis without useful mRNA expression.

## DE subset library construction

Unique 25 bp DNA barcodes were cloned after the stop codon of eGFP, in an AAV ITR-containing plasmid construct containing a self-complementary CAG promoter driving eGFP expression (scCAG-eGFP-Barcode-bghPolyA). *Cap* genes were cloned into an AAV rep/cap plasmid (Addgene #64839) for packaging. Individual AAV variants were then packaged separately with constructs containing different barcodes using a triple transfection method. Variants were then titer matched and mixed in equal ratios before injection into dogs.

## Histology in primates

For histology, both retinas were lightly fixed in 4 % paraformaldehyde, and transferred to PBS. Retinas were then embedded in 5 % agarose and sectioned at 100 μm on a vibratome. Tissue was then examined by confocal microscopy. Antibodies for labeling were anti-GFP (A11122, Thermo, 1:250) and peanut agglutinin (PNA, Molecular Probes, 1:200) a lectin that specifically binds to the cone photoreceptor extracellular matrix.

## Pooling and quantification of scAAVengr libraries

Packaging constructs, containing scCAG-eGFP-Barcode-bghPolyA were constructed as for the DE subset libraries. Approximately equal quantities of AAV serotypes were packaged and pooled. The total titer of pooled virus was: ~ 2.5E + 12–5.0E + 12, see *Supplementary file 1*. Deep sequencing was used to quantify the relative abundance of vector in the pooled library, by amplifying using primer/adapters and sequencing on a MiSeq Nano flow cell. Titers for all variants in the pool were determined

to be within ±1 log from the average variant in the pool, a range that was found to be compatible with accurate normalization across samples.

## Single-cell dissociation of primate retina

The NHP retinas were dissected, and regions of interest were isolated (macula, superior, and inferior periphery). For cynomolgus macaque, superior, and inferior periphery were pooled. Retinal tissue was placed in Hibernate solution (Hibernate A -Ca Solution, BrainBits LLC), and cells were then dissociated using Macs Miltenyi Biotec Neural Tissue Dissociation Kit for postnatal neurons (130-094-802) according to manufacturer's recommendations. Dissected retina pieces were incubated with agitation at 37 °C and further mechanically dissociated. The dissociated neural retina was filtered using a 70 μm MACS Smart Strainer (Miltenyi Biotec) to ensure single-cell suspension. Cells were resuspended in 0.1 % BSA in D-PBS and processed immediately for scRNA-seq.

## Single-cell dissociation of mouse tissues

Brain, heart, and liver of mice were freshly dissected, and cells were dissociated using Macs Miltenyi Adult Brain Tissue Dissociation Kit (130-107-677), Multi Tissue Dissociation Kit 2 (130-110-203) and Liver Dissociation Kit (130-105-807) according to manufacturer's recommendations. The cells were resuspended in 0.1 % BSA in D-PBS and processed immediately for scRNA-seq.

## FACS

Following dissociation using Macs Miltenyi Tissue Dissociation Kits specific for retina, brain, heart, and liver, a Miltenyi MACS Tyto sorter was used to enrich for GFP-positive cells. Cells were resuspended in 0.1 % BSA in D-PBS and processed immediately for scRNA-seq.

## Single-cell RNA-seq of primate retina

Marmoset and cynomolgus macaque samples were prepared for single-cell analysis using a 10x Chromium Single Cell 3' v3 kit. Briefly, single cells from retina samples were captured using a 10x Chromium system (10x Genomics), the cells were partitioned into gel beads-in-emulsion (GEMS), mRNAs were reverse transcribed and cDNAs with 10x Genomics Barcodes were created with unique molecular identifiers (UMIs) for different transcripts. Purified cDNA was PCR amplified and further purified with SPRIselect reagent (Beckman Coulter, B23318). Final libraries were generated after fragmentation, end repair, A-tailing, adaptor ligation, and sample index PCR steps according to 10x Single Cell 3' workflow. An additional targeted sequencing analysis was run on these 10x-prepped cDNA samples, using PCR amplification with Q5 High Fidelity DNA Polymerase to target the GFP sequence and its associated AAV-barcode. 10x libraries were pooled and all samples were submitted for deep sequencing on an Illumina Novaseq S4 flowcell at the UPMC Genome Center. Sequencing depth was targeted at 100,000 reads per sample for the standard scRNA-seq analysis. Sequenced samples were processed and analyzed on Bridges and Bridges-2 through the Extreme Science and Engineering Discovery Environment (XSEDE) (*Towns et al., 2014*). Samples were also analyzed using resources from the University of Pittsburgh Center for Research Computing.

## Single-cell RNA-seq of mouse tissues and cultured cells

Samples from mouse tissues and cultured 293AAV (Cell Biolabs) cells were prepared for single cell analysis using a 10x Chromium Single Cell 3' v3.1 kit. The resulting libraries were pooled, and an additional targeted gene enrichment protocol was performed using 10x Chromium Targeted Gene Expression kit. Samples were submitted for deep sequencing on Illumina Novaseq S2 or SP flow cells.

## Single-cell RNA-seq pre-processing

Sequencing data was demultiplexed into sample-level fastq files using Cell Ranger mkfastq (v3 10x Genomics). Alignment and cell demultiplexing were run using STARsolo (*Dobin et al., 2013*) (v2.7) with default parameters. DropletUtils (*Lun et al., 2019*) (v1.4.3) was used after STARsolo to remove empty droplets (lower.prop = 0.05). Cynomolgus macaque samples were aligned to the Macaca_fascicularis_5.0/macFas5 reference obtained from UCSC and marmoset samples were aligned to ASM275486v1 obtained from Ensembl. Gene annotation for the cynomolgus macaque was created by lifting over the pre-mRNA gene annotations from the hg38 Ensembl human genome. ASM275486v1 gene annotation

files from Ensembl were used for the marmoset. Mouse samples were aligned to the GRCm38 reference GCA_000001635.5 from NCBI and annotated with the GENCODE vM17 basic annotation file.

Cell-free RNA contamination in droplets was estimated using SoupX (*Young and Behjati, 2020*) (v0.3.1). We estimated contamination using genes selected from SoupX's inferNonExpressedGenes method, which identifies genes with highly bimodal expression in the samples. The gene expression in cynomolgus macaque samples was adjusted according to the SoupX estimates, using the 'adjust-Counts' method. No indication of cell-free RNA contamination was observed in marmoset or mouse samples, based off of global expression of key marker genes, and therefore gene expression was not adjusted.

Doublets (10x droplets containing two cells instead of one) were then identified using SCDS (*Bais and Kostka, 2020*) (v1.0.0). Any droplets with a hybrid score >1.3 were considered doublets. Size factor normalization of the single-cell gene expression was achieved using Scran (*Lun et al., 2016*) (v1.12.1), and replicates as well as left/right eyes of the same region were combined for normalization. Finally, imputation strategies were used to denoise the high sparsity that is common in scRNA-sequencing (ALRA v1.0 *Linderman et al., 2018*).

## Single-cell RNA-seq cell identification

Scanpy (*Wolf et al., 2018*) (v1.4.4.post1) was used for the analysis of the scRNA-seq data. First, the top 50 principal components of the gene expression matrix were computed and the Euclidean distance between cells was calculated in this low dimensional space. Then, the distances of 0.5 % of the closest neighbors were kept for each cell and embedded into a neighborhood graph using the UMAP algorithm. Finally, Leiden clustering was performed on the single cell neighborhood graph. Batch correction was performed to combine samples within the same species (including samples across the two marmosets as well as FACS-sorted/non FACS-sorted cynomolgus macaque samples) using Scanorama (*Hie et al., 2019*) (v1.2) and clustering was performed on the batch-corrected values. If samples were batch corrected, normalized counts were saved as raw data and used for differential gene expression analysis.

Cell types were determined by running a differential gene expression analysis using Scanpy's 'rank_gene_groups' function. We used a hypergeometric test and calculated the significance of the intersection of marker genes from one cluster with the published marker genes of each retinal cell type. A Bonferroni p-value correction was applied to account for multiple-hypothesis test. Each cluster is assigned a cell type based on the most significant marker gene intersection p-value. For clusters where the hypergeometric test could not identify a specific cell type match, we annotated the cell type based on marker gene expression using a known cell type marker database. We used two scRNA-seq retina papers (*Macosko et al., 2015*; *Peng et al., 2019*) to construct our database of marker genes for the retina as well as a larger aggregated scRNA-seq marker database (*Zhang et al., 2019*). For the mouse samples taken from other organs, we created marker gene sets using the Tabula Muris dataset (*Schaum, 2018*) as well as information from a combination of organ-specific papers for the brain (*Zeisel et al., 2018*; *Zeisel et al., 2015*) and heart (*Cui et al., 2019*).

## Statistical analysis

Statistical tests were run in R. Normality of the datasets was checked using the Shapiro-Wilk test, and it was found that the datasets were unlikely to be normally distributed (p-values < 2.2E-16 for percent cells infected, p-values < 5.86E-13 for mean transcripts in infected cells). Friedman's test was run on 8 samples from the marmoset and cynomolgus macaque comparing the total percent of all cells infected in each sample across the AAV variants. Additional Friedman's tests were run for each cell type, analyzing the percentage of cells infected across variants on a cell type level. One-sided Wilcoxon signed-rank tests were run on the same datasets (total cells and individual cell types), comparing K912 or NHP26 with the other AAV variants, and the Benjamini-Hochberg method was used to correct p-values. AAV variants were also compared by analyzing the average transcripts in infected cells using the same statistical procedure (*Supplementary files 2 and 3*).

## PCR amplification and enrichment of AAV-barcodes for scAAVengr analysis

The performance of AAV variants was analyzed based on quantification of AAV variant-mediated GFP-barcode mRNA expression (AAV-barcodes). For non-human primates, AAV-barcodes were analyzed

from (1) the original scRNA-seq data and (2) PCR amplification of GFP from the 10x single cell prepped sample library. Mouse samples were analyzed using AAV-barcodes from (1) the original scRNA-seq data and (2) targeted gene enrichment against GFP and other marker genes. Targeted gene enrichment samples from the mouse were downsampled to a similar number of reads as the non-human primate GFP PCR-amplified non-human primate samples.

AAV-barcodes were identified using Salmon (*Patro et al., 2017*) (v0.9.1) transcript quantification. Only reads with one hit to an AAV-barcode were kept. Using these reads, AAV variants were identified based on the AAV-barcode. 10x barcodes in the reads from the PCR amplification analysis were corrected according to the 10x Cell Ranger count algorithm to mitigate any errors that may have been introduced by multiple rounds of PCR. As each UMI (unique molecular identifier) represents a single mRNA molecule captured, only one AAV-barcode should exist for each UMI. Rarely, multiple AAV-barcodes were found per UMI – possibly due to sequencing/PCR-introduced errors – in which case the AAV variant with the highest number of counts for that UMI was kept.

The PCR-amplified barcodes resulted in a higher number of AAV variants found and 10x barcodes with AAV that were identified from the original scRNA-seq data were added to this set. Additionally, AAV variants from the scRNA-seq dataset were added to the set if that 10x barcode was present in the PCR-amplified analysis but that AAV variant was not previously reported.

After identifying AAV variants for each 10x barcode, the 10x barcodes were mapped to the cell types identified previously during the standard scRNA-seq analysis. Once mapped to their respective cell types, AAV counts were normalized by dividing by the total transcriptome nUMI for that 10x barcode and corrected by the dilution factor for each AAV variant. Variants were then divided by the dilution factor of the variant with the highest percentage of cells infected.

## 293AAV infectivity analysis

293AAV (HEK293) cells were downsampled to 300,000 reads and GFP was quantified as previously described using Salmon. 10x cell barcodes were corrected according to the 10x Cell Ranger count algorithm. Cells were not run through any additional quality control steps traditionally used in whole transcriptome single cell analysis, such as empty droplets, as these steps were not applicable to targeted enrichment data. Therefore, corrected 10x barcodes were used for the final cell count and UMI counts originating from GFP were used to estimate the average number of transcripts per cell.

## CRISPR-Cas9 editing analysis

K912 was packaged with an SaCas9 construct (Addgene; pX601-AAV-CMV::NLS-SaCas9-NLS-3xHA-bGHpA;U6::BsaI-sgRNA, Plasmid #61591). The gRNA was designed to target 285 bp downstream of the *RHO* start codon. A cynomolgus macaque and a rhesus macaque were both injected intravitreally, and 9 weeks (cyno) or 6 weeks (rhesus) later, tissue was collected for processing. Genomic DNA was extracted using a Qiagen DNeasy Kit and the target site in the *RHO* gene was PCR amplified with primers attached to Illumina adapter sequences. Amplicon sequences targeting *RHO* were sequenced on an Illumina iSeq and ~1,000,000 reads were recovered for each sample. CRISPResso2 (*Clement et al., 2019*) (v2.0.34) was used to quantify and visualize the edits, using the amplicon sequence and guide sequence as input. Reads were filtered using an average base quality of 30 and single base quality of 20.

## Primers

Primer sequences are listed in *Supplementary file 4*.

## Results

### Directed evolution of AAV vectors in canine retina

In order to develop and validate our pipeline, we first engineered AAV vectors with an enhanced capacity to target the outer retina following intravitreal injection, by implementing directed evolution (DE) of AAV in canines (*Figure 1—figure supplements 1–3*, and methods). DE, which involves applying a selective pressure to libraries of mutated AAV vectors, and conducting iterative rounds of selection, has been used in mouse to create AAV vectors with new abilities to infect Müller glia

(*Klimczak et al., 2009*), to infect photoreceptors (*Dalkara et al., 2013*) and in primates to deliver genes to the outer retina (*Byrne et al., 2020*). Here, we used DE to engineer new AAV vectors with the ability to bypass structural barriers and infect retinal cells following intravitreal injection in canine retinas. Canines are the main preclinical large animal model for development of retinal gene therapies, including the landmark gene therapy clinical trials for *RPE65*-LCA2, due to similar ocular structure and availability of homologous mutant retinal degeneration strains (*Acland et al., 2001*; *Beltran et al., 2012*). Therefore, we hypothesized that canines were a promising model in which to conduct a DE screen.

DE was implemented similarly to the screen previously reported in primate retina (*Figure 1—figure supplement 1 Byrne et al., 2020*). AAV2-based DE libraries, including a ~588 peptide insertion library (which contained a random 7-mer peptide flanked by constant linker sequences LA and A, for a total of 10 amino acids inserted at VP1 position ~588) (*Müller et al., 2003*), an AAV2-Loopswap library (*Koerber et al., 2008*) and an AAV2-ErrorProne library (*Koerber et al., 2006*) were pooled and intravitreally injected into canine eyes (see *Figure 1—figure supplement 2* for a description of each of the AAV libraries and pools used in the study). Promising variants were identified from the DE screen, based on the fold increase over five rounds of selection, normalized to their frequencies in the starting plasmid library. Then, a secondary round of screening in bulk tissue was performed to compare 20 top candidate canine DE variants. These 20 top vectors, along with an AAV2 control, were packaged individually with a ubiquitous CAG promoter driving expression of GFP fused to a unique DNA barcode. Vectors were titer matched, mixed together to create a subset library containing these top-performing DE variants (DE-subset library), and injected intravitreally into both eyes of 3 WT dogs (*Figure 1—figure supplement 3*). Following DNA and mRNA extraction, AAV genome barcodes were PCR amplified from genomic DNA and from cDNA, from photoreceptors and RPE. Variants were ranked on the basis of the normalized change in frequency of their representation in the recovered genomes relative to the injected AAV library (% of total in recovered AAV library / % of total in injected library) (*Figure 1—figure supplement 3*). Rankings based on mRNA and DNA recovery indicated different top-performing variants.

The top ranked variant based on DNA recovery, K916, contains a 10 amino acid insertion (PAPQDTTKKA) at position ~588. The top ranked variant based on mRNA recovery, K912, contains a 10 amino acid insertion (LAPDSTTRSA) at position ~588. The convergent variant from the DE screen, that is, the variant that was most abundant at the end of the screen, K91, which was also overrepresented in the original library, contains a 10 amino acid insertion (LAHQDTTKNA) at position ~588. We have previously shown that convergent variants from DE screens are not necessarily top performers (*Byrne et al., 2020*). That is, greatest fold increase during selection, rather than greatest frequency in the final pool, is the optimal metric for identifying top-performing variants. An additional top-ranking variant based on DNA and RNA recovery, K94, with amino acid insertion ~588 LATTSQNKPA, was also chosen for further testing.

## Single-cell RNA-seq quantification of AAV efficiency across all cell types

Once an initial set of vector candidates has been created, the relative fitness of these variants to infect different types of cells and retinal regions, and to generate abundant transgene expression must be precisely quantified and compared in order to identify optimally efficient vectors and the best candidate vectors for clinical translation. In order to evaluate the performance of AAV variants created through DE in canine retina, and to quantitatively compare their performance to previously engineered AAV variants created through DE in primate retina (*Byrne et al., 2020*) as well as to tyrosine-mutated AAV vectors, we developed a scRNA-Seq based workflow (scAAVengr) (*Figure 1A*). A set of 17 AAV vectors were packaged individually with GFP constructs fused to unique barcodes (the scAAVengr library). Naturally occurring AAV variants included in the scAAVengr library were: AAV1, AAV2 (the parental serotype of the canine and primate DE variants), AAV5, AAV8, AAV9, and AAVrh10. Tyrosine and threonine-mutated versions of AAVs, which have been shown to prevent capsid degradation (*Petrs-Silva et al., 2009*; *Zhong et al., 2008*), included in the scAAVengr library were AAV2-4YF, AAV2-4YFTV, AAV8-2YF, and AAV9-2YF. DE variants included in the set were K91, K912, K916, K94, and primate DE variants NHP9, NHP26, and SCH/NHP26 (*Byrne et al., 2020*). In previous work, NHP9 has been shown to be highly fovea specific. NHP26 has been shown to bypass structural barriers in primate retina at decreased titer (*Byrne et al., 2020*).

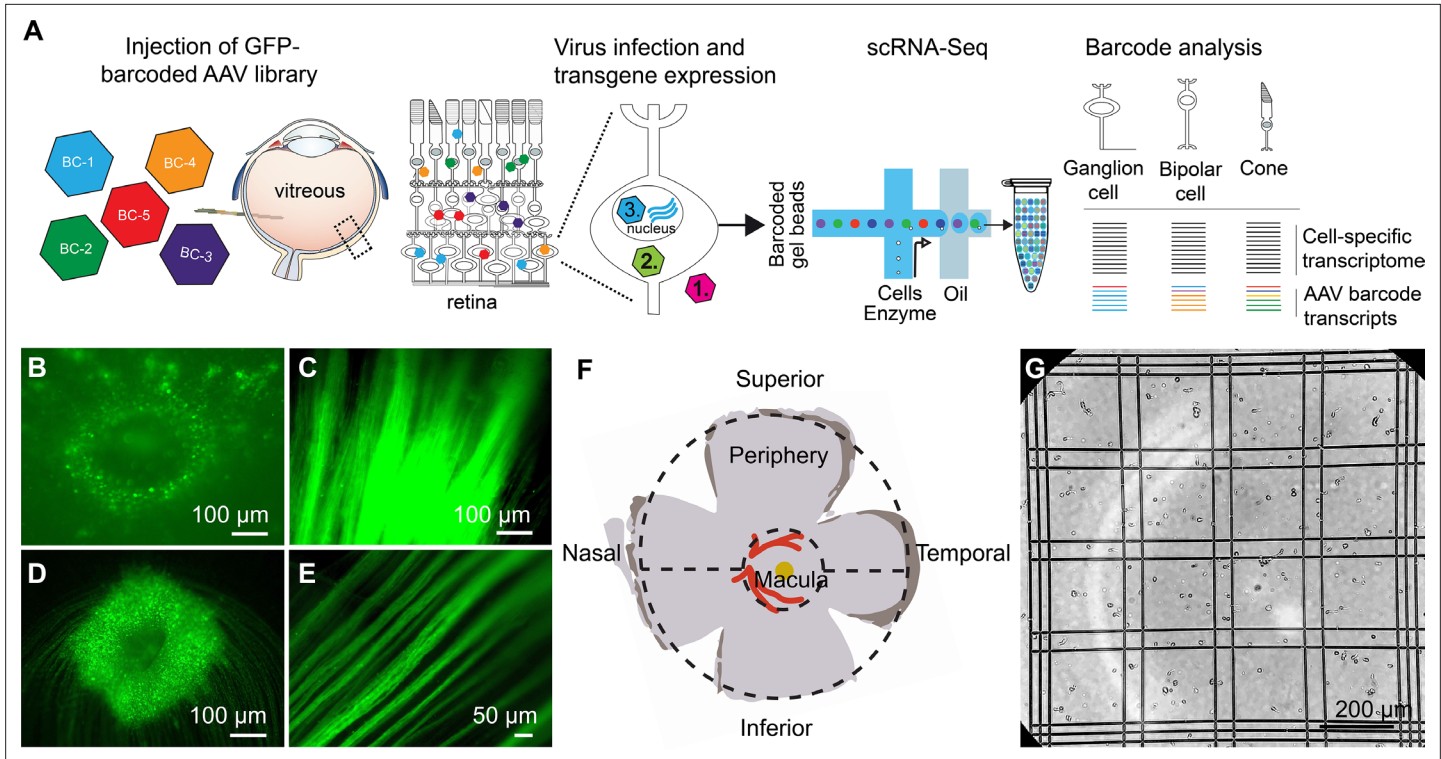

**Figure 1.** scAAVengr pipeline. (**A**) Overview of scAAVengr experimental workflow. An AAV library, consisting of variants packaged with a GFP transgene fused to unique barcodes (AAV-barcodes), was packaged, pooled, quantified by deep sequencing, and injected. Viruses are either noninfectious (1), bind or enter into cells but do not mediate gene expression (2), or traffic to the nucleus resulting in expression of tagged mRNA transcripts (3). Analysis took into account only viruses leading to transgene expression as in (3). Single cell suspensions of the tissue were then created, and a single cell microfluidics system was used to produce single-cell cDNA libraries. Cell types were identified by marker gene expression, and simultaneously, the ability of AAV variants to drive gene expression was evaluated based on quantification of AAV-barcodes in GFP transcripts. (**B–E**) GFP-Barcoded AAV library expression in marmosets and macaques. Intravitreal injection of GFP-barcoded libraries resulted in GFP expression in the retina 8 weeks after injection. (**B**) GFP expression in the perifoveal ring in marmoset retina. (**C**) Axons from retinal ganglion cells in same injected eye as (**B**). (**D**) GFP expression in the perifoveal ring in macaque retina. (**E**) Axons from retinal ganglion cells in same injected eye as (**D**). (**F**) Diagram of primate retinal flatmount. Retinal tissue was collected from macula, and superior and inferior peripheral retina. (**G**) Retinal tissue samples were dissociated into single cell suspensions which were counted using Trypan blue. Trypan blue exclusion is also a test for cell viability. Cell suspensions were then processed through a 10x Chromium scRNA-seq controller.

The online version of this article includes the following figure supplement(s) for figure 1:

**Figure supplement 1.** Directed evolution performed in dogs.

**Figure supplement 2.** Description of libraries used in the study.

**Figure supplement 3.** Injection of GFP-barcoded AAV library of top AAV variants in dogs.

**Figure supplement 4.** Performance of AAV variants following intravitreal injection of DE subset library in dog.

Equal amounts of each GFP-barcoded virus were packaged and pooled. The representation of each variant in the packaged and pooled scAAVengr library was then quantified by deep sequencing. The pooled scAAVengr library was intravitreally injected into the eyes of 3 NHPs (2 marmosets and one cynomolgus macaque, *Figure 1A*, and see *Supplementary file 1*). Eight weeks after intravitreal injection, samples from GFP-expressing retinas were collected (*Figure 1B–E*). Retinal tissue from macula and peripheral regions were dissociated into single cell suspensions (*Figure 1F*), a 10x microfluidics controller was used to create cDNA libraries from single cells (*Figure 1G*), and the cDNA libraries were then sequenced to a depth of 100,000 reads per cell. Raw sequencing reads were aligned (*Dobin et al., 2013*) to the marmoset genome (Ensembl) or the cynomolgus macaque genome (UCSC) and processed with multiple QC methods: empty droplets were identified (*Lun et al., 2019*), ambient RNA was removed (*Young and Behjati, 2020*), doublets were removed (*Bais and Kostka, 2020*), imputation was performed to remove the effects of sparse sampling from sequencing (*Linderman et al., 2018*), single cell gene expression was normalized (*Lun et al., 2016*),

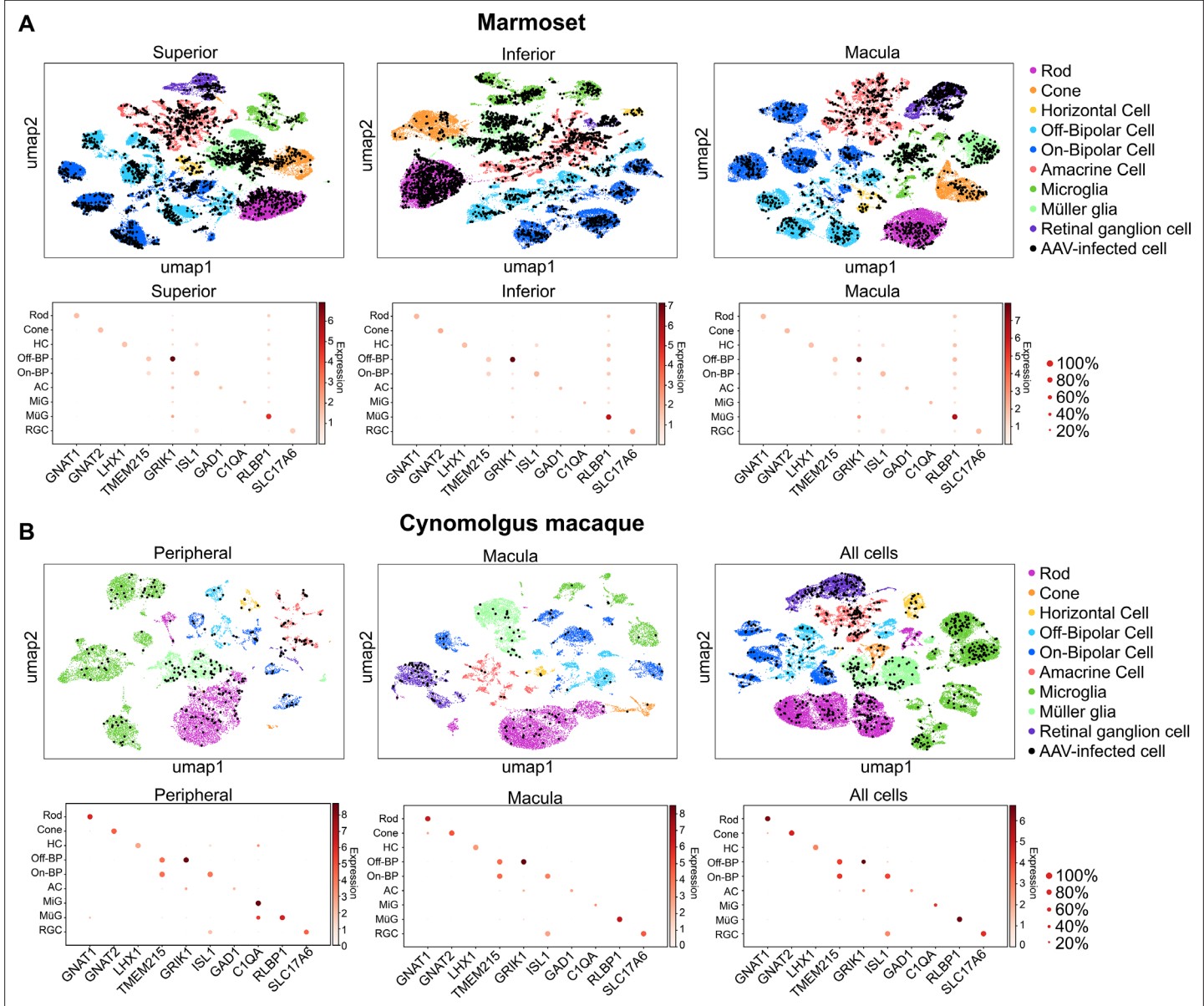

**Figure 2.** Clustering and quantification of AAV-infected retinal cells. (**A**) AAV-infected marmoset retinal cells. Maps of clustered cells from superior, inferior or macular retina show AAV infection. Cells of similar type cluster together. The cell type of each cluster is indicated by color. AAV-infected cells are shown in black. Below each cluster plot, heat maps show the marker genes used to identify cell clusters. The size of the dot indicates the percent of cells in the cluster expressing the marker gene, and the color indicates the level of marker gene expression. Data is pooled from n = 2 marmosets. (**B**) AAV-infected cynomolgus macaque retinal cells. Data is from n = 1 cynomolgus macaque retina, collected from peripheral or macular retina, or from the total pool of retinal cells including GFP+ FACS-sorted cells.

The online version of this article includes the following figure supplement(s) for figure 2:

**Figure supplement 1.** Illustration of barcoding strategies.

and batch correction was performed (*Hie et al., 2019*). Scanpy (*Wolf et al., 2018*) was used in conjunction with the Leiden clustering algorithm to assign individual cells to clusters. The hypergeometric test was then used to quantify the significance of intersection of a clusters' differentially expressed genes with retinal cell type marker genes identified previously (*Macosko et al., 2015*; *Peng et al., 2019*). Each cluster was assigned a cell identity based on the most significant intersection. Clusters of all major retinal cell types were identified in marmosets and macaques, largely in agreement with previous scRNA-seq performed in primate retina (*Peng et al., 2019*). AAV-barcodes were quantified using Salmon (*Patro et al., 2017*) and mapped to identified cell types using in-house

scripts (*Figure 2*, levels of barcoding in scAAVengr analysis are shown in *Figure 2—figure supplement 1*). The number of cells analyzed, after filtering, were: marmoset superior: 69,799; marmoset inferior: 55,941; marmoset macula:65,023; macaque peripheral: 21,904; macaque central: 33,907; macaque all cells: 55,811.

Three metrics were used to compare vector performance across cell types: First, the absolute number of cells infected by each serotype was quantified (*Figure 3—figure supplement 1*). Second, the percent of total cells infected by each serotype was quantified for each major cell type (*Figure 3A*, and *Figure 3—source data 1* -8). Third, within infected cells, the level of transgene expression was evaluated, relative to total transcripts recovered from each cell (*Figure 3B*, and *Figure 3—source data 9* -16). Each of these metrics was corrected by the dilution factors for variants in the injected library, previously determined by deep sequencing. Heat maps of these metrics revealed that variants engineered through DE using canine retinas and primate retinas markedly outperformed AAV2 and AAV2 tyrosine mutants across cell types and in peripheral and macular retina.

Statistical analysis revealed a significant difference in the percent of total cells infected ($p < 0.001$, Friedman's test, and see *Supplementary files 2 and 3*). Of the canine variants, K912 outperformed other engineered serotypes, in agreement with the results observed in bulk analysis performed in dog retina (*Figure 1—figure supplement 4*). The convergent variant (K91) did not outperform parental serotypes, underscoring the need for deep sequencing to determine top performing AAV variants from the DE screen. Of the primate variants, NHP26 outperformed other variants, infecting major retinal cell types in inner and outer retina in marmoset and cynomolgus macaque retina.

Evaluation of AAV infectivity at cell-type resolution revealed that newly engineered K9 variant AAVs and NHP26 infected inner and outer retinal cells in primate retina (*Figure 3*). Infectivity, in terms of percent cells infected, was most efficient in RGCs and Müller glia, particularly in the macula where the inner limiting membrane is less of an anatomical barrier. In the outer retina, rods and cones were also infected. Higher rates of infection and expression levels were seen in marmosets compared to the cynomolgus macaque.

In order to rank best performing pan-retinal variants and the best performing variants by cell type, variants were plotted by the mean transcripts per cell in infected cells vs. the percent cells infected for each AAV serotype (*Figure 4*). Plots were created with data from all cell types on the same plot (*Figure 4A*) or in individual plots per cell type, for each region tested in each primate (*Figure 4B*.). These plots revealed that K912 was the overall best performing canine variant across retinal cell types, and NHP26 was the top performing primate-derived variant across cell types. Of all the variants tested, K912 was the top performer across cell types.

In order to determine the number of AAV variants infecting a single cell, upset plots were created to show the number and serotypes of AAV particles infecting individual cells. Upset plots show the number of cells infected by a particular combination of AAVs (the intersection size) as well as the number of cells infected by a particular serotype (the set size). The majority of infected cells were infected by a single variant (K912), although many cells were infected by multiple serotypes (*Figure 5A*). As many as eight serotypes infected a single cell in marmoset retina, while up to three serotypes infected a single cell in macaque retina.

Next, in order to further interrogate the dynamics of infection in the context of pooled libraries of AAV variants, and to determine whether the presence of other AAV variants impedes infection of library members, HEK293 cells grown in vitro were infected with either AAV2 alone, or with a pool containing 4 AAV's (AAV2, AAV8, AAV9 and K912), or with a pool containing 16 AAV's (AAV2, AAV8, AAV9, K912, AAV1, AAVrh10, AAV2-4YF, AAV2-4YF-T491V, AAV8-2YF, AAV9-2YF, K91, K916, K94, NHP9, NHP26, and SCH NHP9/26) (*Figure 5B*). 1E + 6 HEK293 cells were infected with (a) AAV2 (MOI of ~6E + 3, 2 technical replicates were performed), or (b) AAV2 (MOI of ~6E + 3)+ AAV8 (MOI of ~4E + 4), AAV9 (MOI ~ 2E + 4) and K912 (MOI ~ 4E + 3), or (c) a pool of 16 variants (total combined MOI of the pool ~5E + 3). In all three conditions (alone, in the presence of three additional variants, or infected in the presence of 16 additional variants), the number of cells infected by AAV2, and the average number of transcripts recovered from infected cells (averaged across all cells infected) were stable. For K912, the number of cells infected and the average number of transcripts recovered from infected cells were stable between the 4-member pool and the 16-member pool. Together these results indicate that competition for receptors, or the presence of additional variants in the library does not impact quantification of AAV performance.

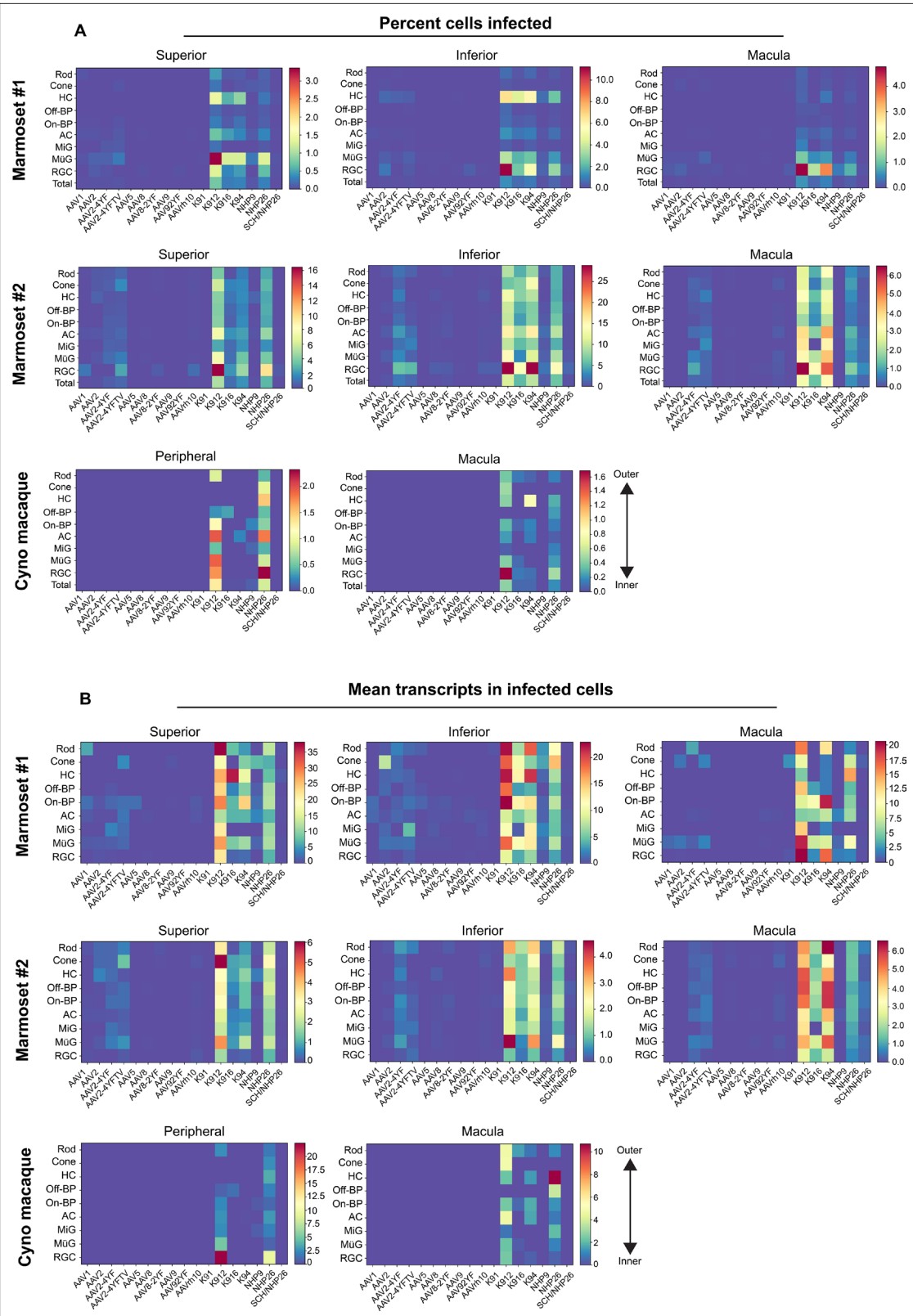

**Figure 3.** Quantitative comparison of variant infection across retinal cell types. (**A**) Percent of cells infected by AAV serotypes in marmoset and cynomolgus macaque retina. Heat maps show the percent of identified cells infected by each serotype in the screen, corrected by the AAV dilution factor, for each retinal cell type. Total = the percent cells infected from the total pool of identified cells. Data is shown for each primate analyzed, across superior, inferior and macular retina. (**B**) Level of expression in infected cells. The mean level of GFP-barcoded transcript expression in cells infected with

*Figure 3 continued on next page*

*Figure 3 continued*

AAV is shown in heatmaps, for all retinal cell types. Cell types are arranged from outermost cells in the retina (rods and cones) to innermost cells (RGCs). Data is averaged across all infected cells and corrected by the AAV dilution factor. Data is shown as mean transcripts per cell/100,000 transcripts. HC = Horizontal Cell; Off-BP = Off-Bipolar Cell; On-BP = On-Bipolar cell; AC = Amacrine Cell; MiG = Microglia; MG = Müller Glia; RGC = Retinal Ganglion Cell.

The online version of this article includes the following figure supplement(s) for figure 3:

**Source data 1.** Marmoset 1-Superior-Percent cells.

**Source data 2.** Marmoset 1-Inferior-Percent cells.

**Source data 3.** Marmoset 1-Macula cells.

**Source data 4.** Marmoset 1-Superior-Transcripts.

**Source data 5.** Marmoset 1-Inferior-Transcripts.

**Source data 6.** Marmoset 1-Macula-Transcripts.

**Source data 7.** Marmoset 2-Superior-Percent cells.

**Source data 8.** Marmoset 2-Inferior-Percent cells.

**Source data 9.** Marmoset 2-Macula cells.

**Source data 10.** Marmoset 2-Superior-Transcripts.

**Source data 11.** Marmoset 2-Inferior-Transcripts.

**Source data 12.** Marmoset 2-Macula-Transcripts.

**Source data 13.** Cyno-Peripheral-Percent cells.

**Source data 14.** Cyno-Macula cells.

**Source data 15.** Cyno-Peripheral-Transcripts.

**Source data 16.** Cyno-Macula-Transcripts.

**Figure supplement 1.** Numbers of AAV-infected cells.

Then, we estimated the number of unique AAV variants that could be directly compared through the scAAVengr pipeline. AAV constructs were cloned (as in *Figure 1—figure supplement 1*, Step 8), containing a CAG promoter driving expression of GFP, which was then fused to a barcode either 3,6,9, or 14 base pairs in length. Each possible nucleotide was equally represented at each position of the barcodes (hand mixed, IDT), with a maximum possible diversity of 64 (3 bp barcode), 4096 (6 bp barcode), 262,144 (9 bp barcode), or 268,435,456 (14 bp barcode) unique barcodes (*Figure 2—figure supplement 1*). These constructs were then packaged into AAV2 and used to infect ~1E + 6 HEK293 cells in vitro at an MOI of 1000. Following onset of GFP expression, 8000 cells were processed through the scAAVengr pipeline, and the number of unique AAV-barcodes recovered was quantified. The number of barcodes recovered were 64 (3 bp barcode), 4096 (6 bp barcode), 109,701 (9 bp barcode), or 78,307 (14 bp barcode), indicating that> E + 5 unique AAV variants could be quantified simultaneously, from a single sample containing 8000 cells.

## Validation of K912 in primate retina

Retinal cell expression with K912, the overall top performer, was then individually validated by packaging and intravitreally injecting a self-complementary CAG-GFP construct in two primates (*Figure 6A–G*, *Figure 6—figure supplement 1*). Ten weeks after injection, GFP expression was evident in retinal flat-mounts and cross sections. Confocal microscopy imaging of PNA (which labels cone inner segments)-labeled peripheral retina, imaged at the level of the photoreceptor layer, revealed GFP expression in rods and cones, which was higher in rods than in cones, in agreement with scAAVengr heat maps. Cross sections showed strong expression in RGCs and Müller glia, which was more efficient than in outer retina, particularly in the macula, also in agreement with scAAVengr heat maps.

Then, as a functional test performed in a therapeutic context, K912 was also packaged with SaCas9 (Addgene #61591) driven by a ubiquitous CMV promoter and a guide RNA targeting rhodopsin, packaged in a single vector (*Figure 6H–J*). This vector was injected intravitreally in a cynomolgus macaque and a rhesus macaque. RT-PCR amplifying SaCas9 cDNA showed expression of Cas9 in injected, but not in uninjected primate retinas (*Figure 6H*). We then used deep sequencing to quantify editing from retinal punches containing all cell types, which revealed that 1.8 % of reads mapping to

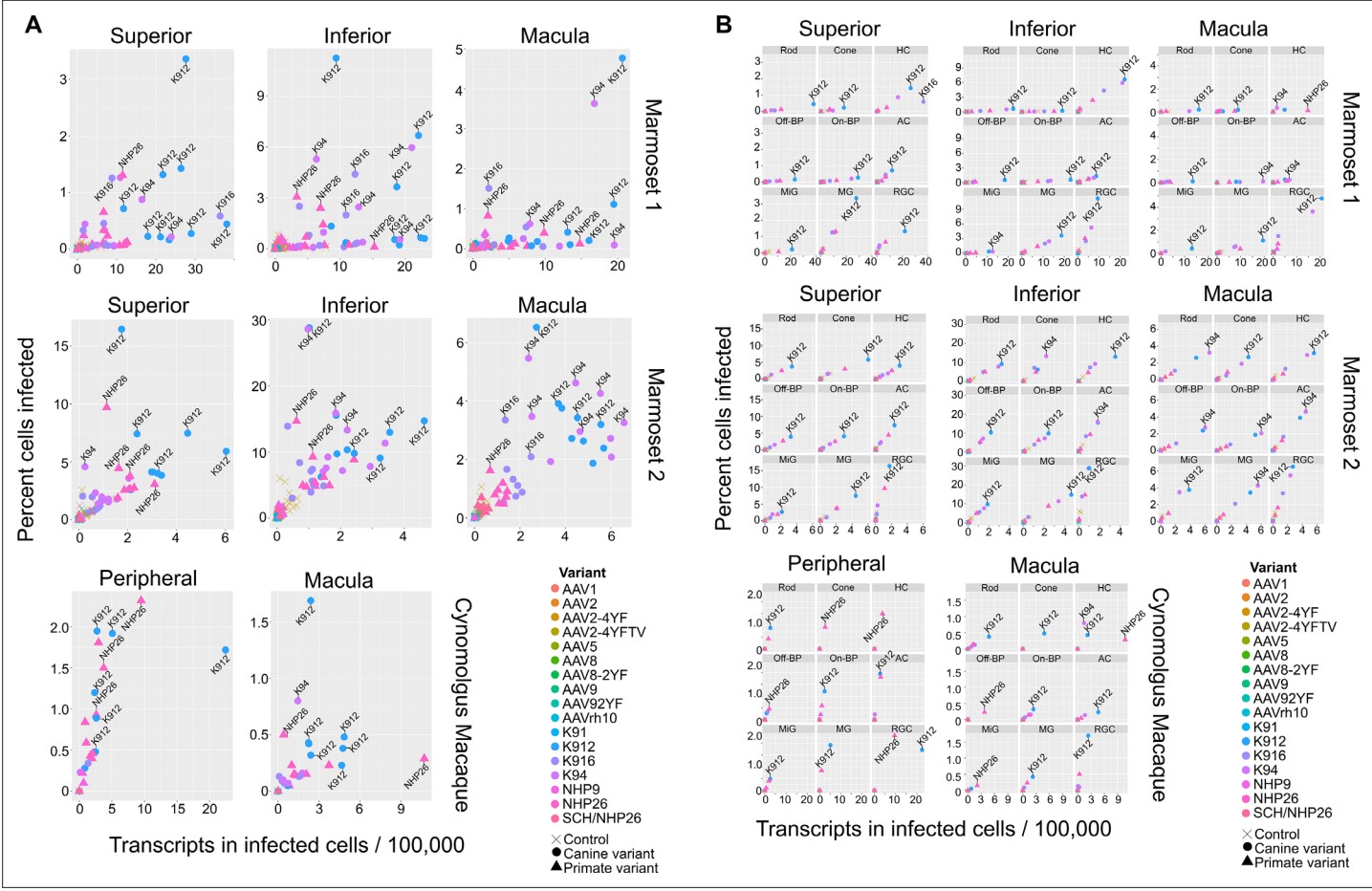

**Figure 4.** Serotype performance across retinal regions. (**A**) Scatter plots reveal that K912 is the overall best performing canine variant across retinal regions, while NHP26 is the best performing primate DE variant. Plots show the number of transcripts in infected cells per 100,000 transcripts vs the percent of cells infected for each serotype. Nine data points are plotted for each variant, one data point for each cell type. Serotypes are indicated by color. Data is from n = 2 marmosets and n = 1 cynomolgus macaque. A subset of the top performing variants, according to each variable, are labeled. Control vectors are shown as X's. Canine-derived variants are shown as circles. Primate-derived variants are shown as triangles. Best performing variants appear toward the upper right hand corner of each plot. (**B**) AAV variant performance in each cell type. Scatter plots reveal that K912 is the overall best performing variant across most retinal cell types and across retinal regions, while NHP26 is the best performing primate-derived variant. Plots show the number of transcripts in infected cells per 100,000 transcripts vs the percent of cells infected. Individual plots show the performance for AAV serotypes (in different colors) in individual cell types, across retinal regions. A subset of the top performing variants, according to each variable, are labeled. HC = Horizontal Cell; Off-BP = Off-Bipolar Cell; On-BP = On-Bipolar cell; AC = Amacrine Cell; MiG = Microglia; MG = Müller Glia; RGC = Retinal Ganglion Cell.

the targeted site showed editing events in the cynomolgus macaque and 1.7 % editing in the rhesus macaque (*Figure 6I and J*), similar to the percent total of K912- GFP infected cells. Individual reads revealed deletions, insertions and base substitutions in the cynomolgus macaque and rhesus macaque following injection with K912-saCas9-gRNA-RHO.

## Validation of scAAVengr pipeline in additional organs and species

Finally, in order to validate the scAAVengr pipeline in other species and tissues, we screened the same 17-member scAAVengr AAV library in mouse brain, heart and liver following systemic injections (*Figure 7*). AAV library was packaged, containing each GFP-barcoded virus, and 50 μL of a 5e + 12 vg/mL titer library was injected via facial vein in P0 mice. The representation of each variant in the packaged and pooled library was quantified by deep sequencing. Three weeks after injection, brain, heart and liver were collected and dissociated into single cell suspensions, and a 10x microfluidics controller was used to create cDNA libraries from single cells. GFP+ cells were enriched using FACS, and AAV infection was quantified across all cell types. cDNA libraries were sequenced to a depth of

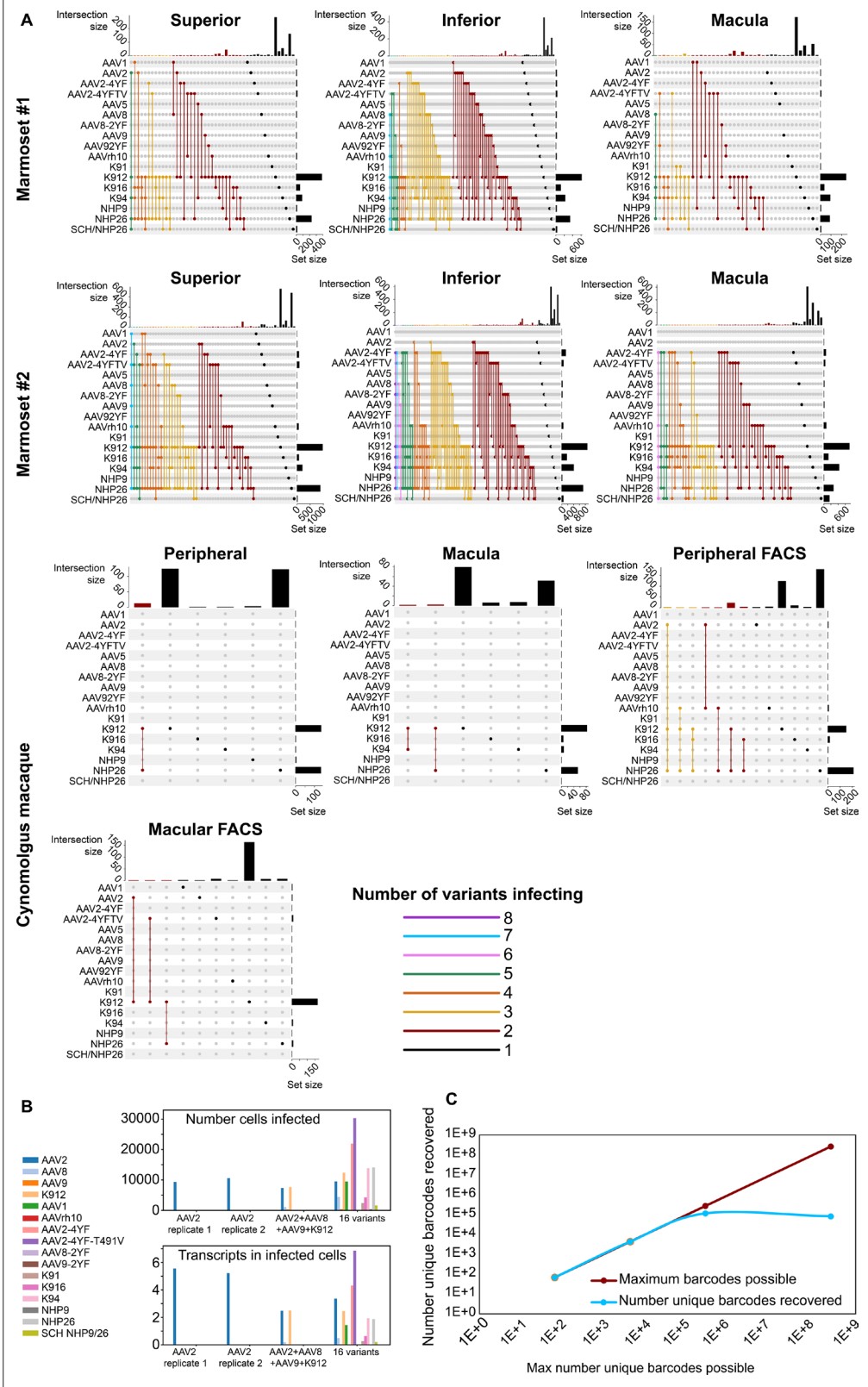

**Figure 5.** Dynamics of infection from multiple AAV serotypes. (**A**) Upset plots show that multiple AAV serotypes can infect the same retinal cell, although the majority of retinal cells are infected by the top performing variants. Plots are shown across multiple regions in marmoset and cynomolgus macaque retina. Dots and connecting vertical lines indicate the serotype and number of variants infecting single cells. The number of cells infected by

*Figure 5 continued on next page*

*Figure 5 continued*

a particular combination of AAVs (the intersection size) is illustrated in the bar graph across the top of the plot. The number of cells infected by a particular serotype (the set size) is shown across the right-hand Y-axis. Lines are colored according to the number of AAV variants in the subset. (**B**) scAAVengr screening performed in HEK293 cells shows that pooling of AAV variants does not impede infection of other library members, either in terms of the number of cells infected or expression levels. AAV2, or a 4-member pool of AAV's, or a 16-member pool of AAV's was used to infect HEK293 cells. The scAAVengr pipeline was then used to quantify infectivity. In all preparations, AAV2 infected similar numbers of cells, and similar levels of expression were observed. K912 infected similar numbers of cells, and similar expression levels were observed in both the four variant and the 16 variant pools. (**C**) At least~ E + 5 AAV variants can be quantified using the scAAVengr pipeline. Libraries containing AAV2, packaged with barcodes of 3,6,9 or 14 base pairs in length (with a maximum possible diversity of 64, 4,096, 262,144, or 268,435,456), were used to infect HEK293 cells. scAAVengr was then used to quantify the number of barcodes recovered. From samples containing 8000 cells, all possible barcodes were recovered from libraries with 3- and 6 bp barcodes. ~ 100,000 unique barcodes were recovered from 9- and 14 bp barcodes, indicating that at least E + 5 variants could be quantified using the scAAVengr pipeline.

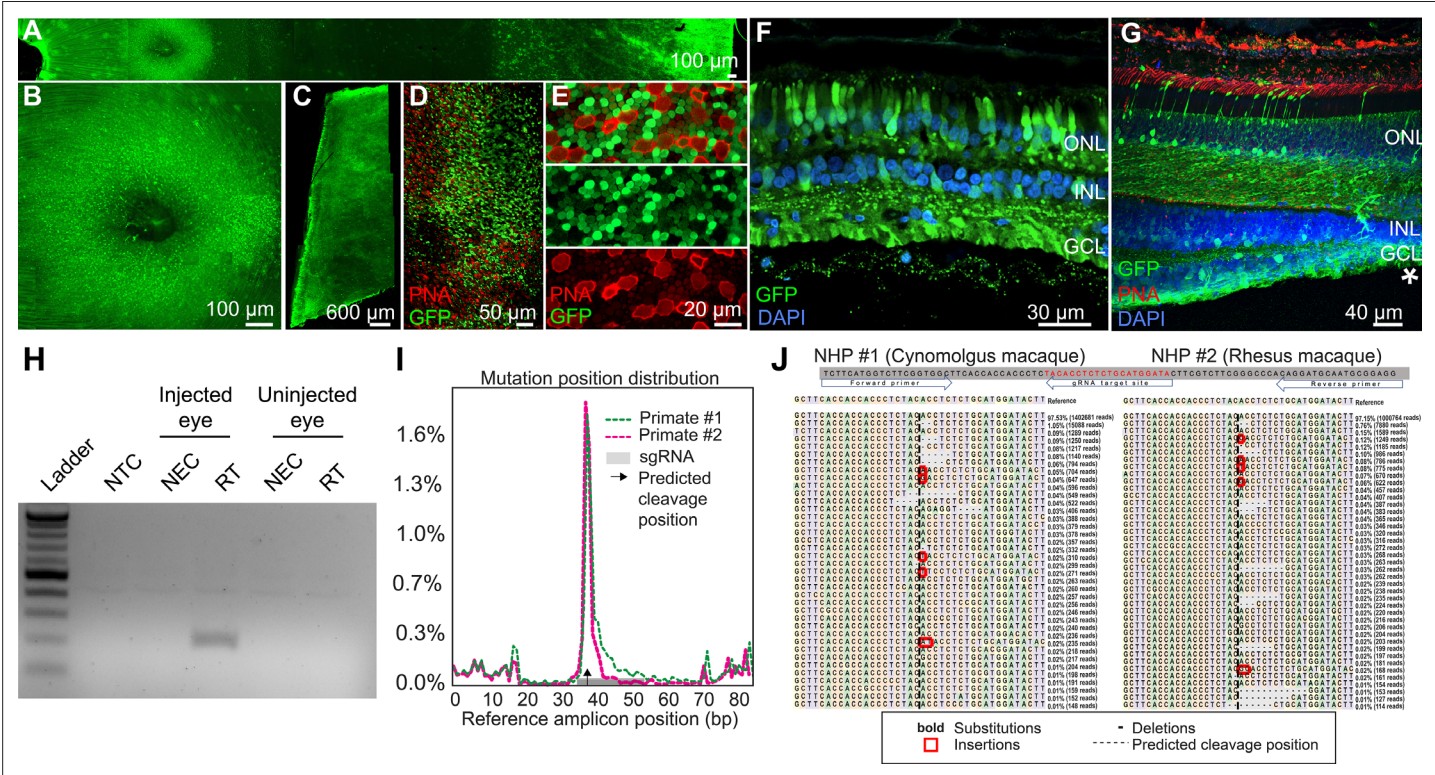

**Figure 6.** K912 expression in primate retina. (**A–G**) GFP expression in a cynomolgus macaque injected with ~2.6E + 12 vg of K912-scCAG-GFP. (**A**) GFP expression in a flatmounted cynomolgus macaque retina 2.5 months after injection. (**B**) GFP expression in the perifoveal ring. (**C**) GFP expression in peripheral retina. (**D**) Flatmount imaged through the photoreceptor layer in peripheral retina showing GFP expression and PNA labeling of cones. (**E**) Higher resolution image of peripheral photoreceptors, labeled with PNA. (**F**) Cross section of peripheral retina showing GFP expression and DAPI labeling of nuclei. (**G**) Cross-section through the foveal edge showing GFP expression. Cone outer segments are labeled with PNA. Nuclei are labeled with DAPI. (**H**) RT-PCR of cDNA from injected and uninjected eyes. RT-PCR shows Cas9 expression in macula of the cynomolgus macaque retina injected with K912-saCas9-gRNA-*RHO* but not in a control uninjected eye. (**I**) Percent of genome editing and location of editing relative to guide RNA sequence in two macaques injected with K912-scCAG-saCA9-gRNA-*RHO*. (**J**) Deep sequencing reads showing deletions, insertions and base substitutions in the cynomolgus macaque and rhesus macaque following injection with K912-saCas9-gRNA-*RHO*.

The online version of this article includes the following figure supplement(s) for figure 6:

**Figure supplement 1.** Expression of K912-CAG-GFP in cynomolgus macaque retina.

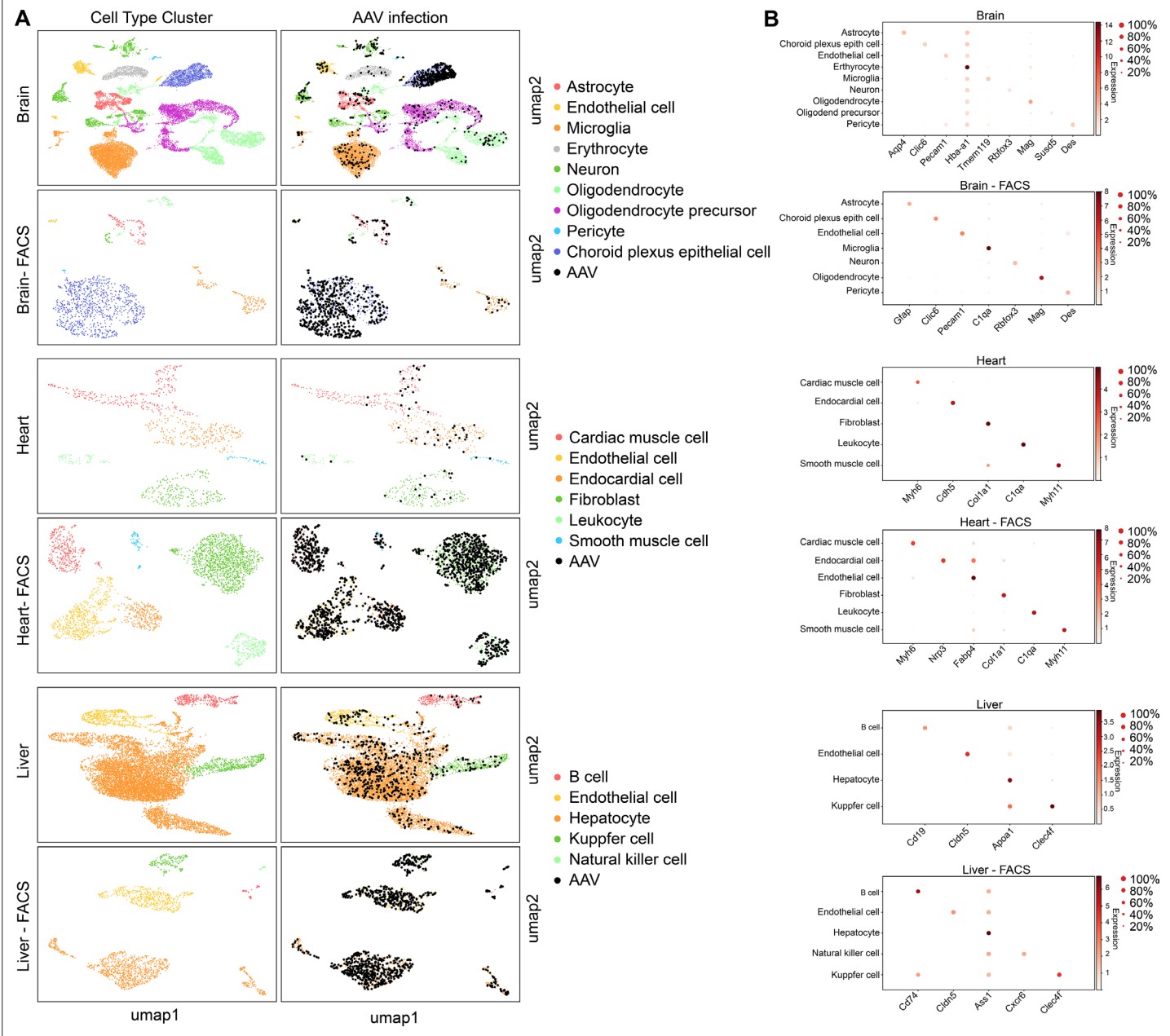

**Figure 7.** scAAVengr quantified the performance of AAV variants in mouse brain, heart, and liver following systemic injection of libraries. (**A**) Maps of clustered AAV-infected cells from brain, heart, and liver. The cell type of each cluster is indicated by color. AAV-infected cells are shown in black. Maps are shown for cells processed directly following single-cell dissociation, or following FACS sorting to enrich for GFP+ cells. (**B**) Heat maps show the marker genes used to identify cell clusters in each sample. The size of the dot indicates the percent of cells in the cluster expressing the marker gene, and the color indicates the level of marker gene expression. Data is pooled from n = 2 mice.

100,000 reads per cell. Raw sequencing reads were aligned to GRCm38. Reads were processed with QC methods as previously described in order to identify empty droplets (**Lun et al., 2019**), remove doublets (**Bais and Kostka, 2020**), perform imputation to remove the effects of sparse sampling from sequencing (**Linderman et al., 2018**) and normalize single cell gene expression (**Lun et al., 2016**). Scanpy (**Wolf et al., 2018**) was used in conjunction with the Leiden clustering algorithm to assign individual cells to clusters. The hypergeometric test was then used to quantify the significance of intersection of a clusters' differentially expressed genes with cell type marker genes identified previously (**Cui et al., 2019**; **Schaum, 2018**; **Zeisel et al., 2018**; **Zeisel et al., 2015**). Each cluster was assigned

a cell identity based on the most significant intersection. The number of cells analyzed, after filtering, were: brain: 17,373; brain FACS: 1,213; heart: 980; heart FACS: 2,746; liver: 10,397; liver FACS: 1,688.

The absolute number of cells infected by each serotype was quantified. Then, the percent of total cells infected by each serotype was quantified for each major cell type and used to create heat maps of serotype infectivity (*Figure 8A*, and *Figure 8—source data 1* -6). Finally, within infected cells, the level of transgene expression was evaluated, relative to total transcripts recovered from each cell (*Figure 8B*, and *Figure 8—source data 7* -12). Each of these metrics was corrected by the percent total of each of the variants in the injected library, previously determined by deep sequencing. Heat maps of these metrics revealed that variants AAV8, AAV8-2YF, AAV9, AAV9-2YF, and AAVrh.10 infected brain, heart, and liver following neonatal systemic injections. AAV1 and AAV5 also infected liver cells. These results are in agreement with previously published data on the tropism of these AAV serotypes (*Duan, 2016*; *Foust et al., 2009*; *Wang et al., 2010*; *Wang et al., 2005*; *Yang et al., 2014*; *Zhang et al., 2011*; *Zincarelli et al., 2008*). SCH/NHP26, a variant created through DE in primate with a backbone partially based on AAV9, also infected brain, heart and liver. In contrast, AAV2-based retinal DE variants, including K912 and NHP9, did not efficiently infect organs outside of the eye.

## Discussion

Together these results validate the scAAVengr pipeline as a platform for simultaneous quantitative evaluation and ranking of new AAV serotypes with cell-type resolution. Quantitative comparisons of newly engineered vectors, including evaluation of transgene expression levels and cell-type tropism, have, in the past, required large numbers of animals. In primates, such studies have been impractical, due to the associated ethical and financial burden. Additionally, in primates, the large variability between animals has made comparisons between animals inaccurate, and evaluation of multiple AAVs in the same animal was not possible. In order to address these problems, we developed a single-cell RNA-seq AAV engineering (scAAVengr) pipeline for quantitative head-to-head in vivo comparison of transgene expression from newly engineered AAV capsid variants. By simultaneously quantifying cellular and viral RNA at the single-cell level, this method allows for efficient, direct, and head-to-head comparison of multiple vectors across all cell types in the same animals. Pan-tissue efficiency of transgene expression can be determined, as well as specificity for any cell-type of interest that can be identified by its transcriptome profile. The number and identity of unique AAV serotypes infecting a single cell can be observed, and the efficiency and specificity of a potential gene therapy can be accurately estimated.

Here, we have evaluated the AAV tropism of newly engineered AAV capsid variants, created in the context of dog retina, with increased ability to infect all major retinal cell types. These variants were directly compared to variants created through DE in primate eyes, as well as naturally occurring variants and tyrosine modified versions. The overall top-performing AAV, K912, was identified through screening performed in the context of the canine retina. Canines, which have an even thicker vitreous than primates, may represent a more difficult model, with a higher barrier for transduction, resulting in better performing viruses. Further work will need to be done to determine whether vectors from canine screens consistently out-perform variants screened in other animal models.

The overall rate of infection achieved by K912, around 2 % of total retinal cells, suggests that significant improvements in AAV transduction must be achieved for maximal therapeutic benefit to be achieved via intravitreal injections. Additional improvements to AAV screening protocols or library construction may be required in order to find even better performing vectors. Here, the best performing variants were determined by using the cloned AAV DE library (plasmid used for packaging) as the denominator for evaluation. Additional work will be required to determine whether the packaged AAV DE library (Round 0) may be a better common denominator to follow DE enrichment. It remains to be seen whether barriers to retinal penetration by AAV vectors from intravitreal injection can be overcome by modification of the AAV capsid alone, or whether alternative methods for vector delivery are required. The scAAVengr pipeline may also be useful in determining absolute rates of infection and direct comparison with other routes of administration or methods for transgene expression.

The 17-member scAAVengr library of AAV variants was also analyzed in brain, heart and liver following systemic injection in neonatal mice. Variants including K912 and NHP9, which performed well in primate retina, did not infect mouse brain, liver and heart, which indicates that the mutations

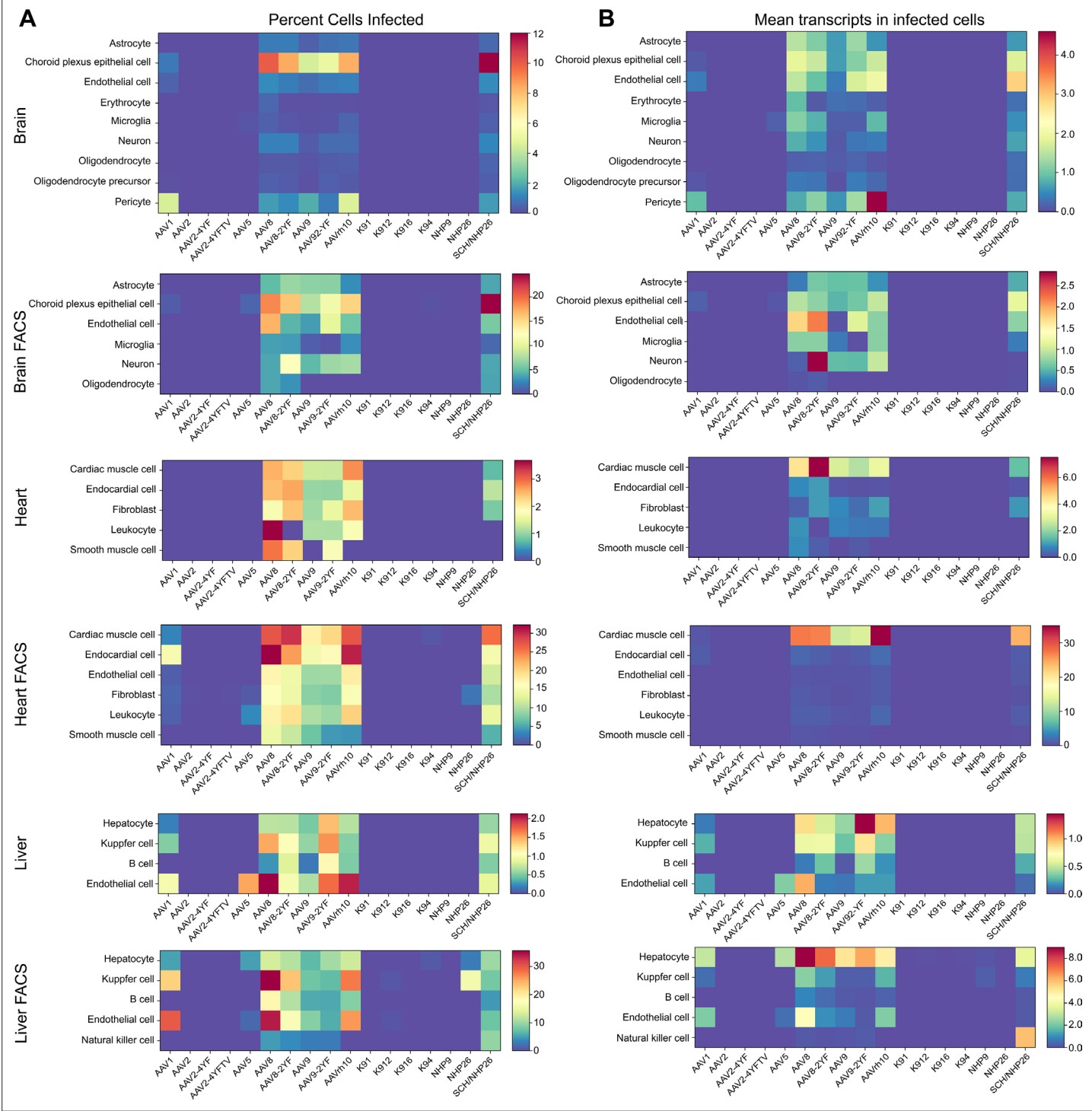

**Figure 8.** Quantitative comparison of variant infection across cell types in mouse brain, heart and liver. (**A**) Percent of cells infected by AAV serotypes. Heat maps show the percent of identified cells infected by each serotype in the screen, corrected by the AAV dilution factor, for each retinal cell type. Data are from the tissues of 2 mice. (**B**) Level of expression in infected cells. The mean level of GFP-barcode transcript expression in cells infected with AAV is shown in heatmaps, for all identified cell types. Data is averaged across all infected cells and corrected by the AAV dilution factor. Data is shown as mean transcripts per cell/100,000 transcripts.

The online version of this article includes the following figure supplement(s) for figure 8:

**Source data 1.** Brain-Percent cells.

**Source data 2.** Brain-Transcripts.

*Figure 8 continued on next page*

*Figure 8 continued*

**Source data 3.** Brain FACS-Percent cells.

**Source data 4.** Brain FACS-Transcripts.

**Source data 5.** Heart-Percent cells.

**Source data 6.** Heart-Transcripts.

**Source data 7.** Heart FACS-Percent cells.

**Source data 8.** Heart FACS-Transcripts.

**Source data 9.** Liver-Percent cells.

**Source data 10.** Liver-Transcripts.

**Source data 11.** Liver FACS-Percent cells.

**Source data 12.** Liver FACS-Transcripts.

which confer increased ability to cross structural barriers in the retina are not sufficient for infection from systemic injection in neonatal mice. Further analysis could be performed, using the same dataset, to quantify infectivity in specific subtypes of retinal cell types, as subtype marker genes are validated. Additional work is required to determine the maximum number of pooled AAV variants that can be screened simultaneously, though our results indicate that libraries containing at least E + 5 variants can be evaluated from a single sample containing 8000 cells. Quantification revealed that multiple AAV serotypes do infect individual cells, suggesting that competition between variants does not inhibit infection. Analysis of in vitro infection of AAV variants in HEK293 cells also suggests that competition between variants infecting the same cells does not significantly affect the number of infected cells or the level of transgene expression in infected cells, validating infectivity data from pooled AAV variants. However, in vivo confirmation of this finding is warranted, as the barriers affecting the dynamics of infection may differ in a more complex environment, such as the retina.

Marmoset eyes, which are significantly smaller than human and macaque eyes, were more easily infected than cynomolgus macaques, with greater percentages of cells infected and more infection events per cell. Marmoset eyes may represent a less challenging target for AAV transduction than cynomolgus macaques due to more efficient diffusion of viral particles in smaller eyes, a less rigid vitreous consistency, or immunological factors. This suggests that large primates may be of more use for accurate prediction of performance of gene delivery efficiencies in humans, particularly in the retina, but potentially in other targets as well, such as infection of organs from systemic injections, where diffusion rates and structural barriers may differ between marmosets and macaques. The rate of Cas9 editing in macaques was similar to the total percent of cells infected by K912, indicating that editing was efficient in infected cells. Importantly, similar rates of infection and genome editing indicate that scRNA-seq can accurately estimate the efficiency of viral gene delivery.

Data from primate retina and mouse brain, heart, and liver validate that the scAAVengr workflow is applicable to any tissue for which cell type marker genes are available, and provides a rapid, quantitative method by which AAV vectors can be rapidly evaluated for their clinical potential. This method enables the definitive ranking of AAV's, in terms of transgene expression efficiency, and in future studies will enable the identification of vectors with greater cell-type specificity. The quantitative nature and single-cell resolution provided by the scAAVengr pipeline therefore enables identification and development of optimal AAV vectors for clinical translation.

## Acknowledgements

General: Deep sequencing was performed at the UC Berkeley Vincent J Coates sequencing facility and the UPMC Genome Center. Confocal imaging of canine and primate retinas was performed at the Berkeley Biological Imaging Facility. At the University of California Berkeley we thank Tim Day, Yvonne Lin, Cécile Fortuny, and Jaskiran Mann for technical advice and assistance. At the University of Pennsylvania, we thank Lydia Melnyk for research coordination, and the staff at the Retinal Diseases Study facility (University of Pennsylvania) for animal care. At the University of Pittsburgh, we thank DLAR staff for animal care. We thank Peter Strick for breeding of marmosets. This work used the Extreme Science and Engineering Discovery Environment (XSEDE) (Towns et al., 2014), which is supported by National Science Foundation grant number ACI-1548562. Specifically, it used the Bridges and

Bridges-2 system, which is supported by NSF award number ACI-1445606 and ACI-1928147, at the Pittsburgh Supercomputing Center (PSC). This research was also supported in part by the University of Pittsburgh Center for Research Computing through the resources provided.

## Additional information

### Competing interests

Meike Visel: is an inventor on AAV capsid variants (US patent IDs: 10,214,785, 10,745,453). MV has also received royalty payments from UC Berkeley. The author has no other competing interests to declare. José-Alain Sahel: has served as a consultant (with no consulting fee) for Pixium Vision, GenSight Biologics and SparingVision. Personal financial interests: Pixium Vision, GenSight Biologics, Prophesee and Chronolife, SparingVision, SHARPEYE, Vegavect, Newsight Therapeutics. The author has no other competing interests to declare. David V Schaffer: is named as an inventor on patent applications on AAV capsid variants (U.S. Patent Applications No. 16/315,032, 16/486,681). DS is also a co-founder of 4D Molecular Therapeutics, and DS performs consultancy and owns stock options in this company. The author has no other competing interests to declare. Andreas R Pfenning: has received an honorarium from the University of Rhode Island, and has applied for patents on specific Nuclear-Anchored Independent Labeling System (PCT/US2020/038520 and PCT/US2020/038528). The author has no other competing interests to declare. John G Flannery: is an inventor on patent application on AAV capsid variants (U.S. Patent Application No. 16/315,032, 16/486,681). The author has no other competing interests to declare. William A Beltran: is an inventor on patent application on AAV capsid variants(16/315,032). The author has no other competing interests to declare. William R Stauffer: is an inventor on a patent application for methods of AAV capsid development (PCT/US2019/068489). The author has no other competing interests to declare. Leah C Byrne: is named as an inventor on patent applications on AAV capsid variants and AAV screening methods (U.S. Patent Applications No. 16/315,032, 16/486,681, PCT/US2019/068489). LB has consulted on AAV-mediated gene therapy for Vedere Therapeutics, and is a named founder of Vegavect and Newsight Therapeutics. The author has no other competing interests to declare. The other authors declare that no competing interests exist.

### Funding

| Funder | Grant reference number | Author |
| --- | --- | --- |
| Ford Foundation | | Leah C Byrne |
| National Eye Institute | F32EY023891 | Leah C Byrne |
| National Eye Institute | R24EY-022012 | David V Schaffer<br>John G Flannery<br>William A Beltran |
| National Eye Institute | R01EY017549 | Gustavo D Aguirre<br>William A Beltran |
| National Eye Institute | P30EY001583 | Gustavo D Aguirre<br>William A Beltran |
| National Institute of Mental Health | UG3MH120094 | Andreas R Pfenning<br>Leah C Byrne<br>William R Stauffer |
| National Institute of Mental Health | DP2MH113095 | William R Stauffer |
| Research to Prevent Blindness | Career Development Award | Leah C Byrne |
| Foundation Fighting Blindness | Individual Investigator Research Award | Leah C Byrne |
| UPMC Immune Transplant and Therapy Center | | Leah C Byrne |

| Funder | Grant reference number | Author |
|---|---|---|
| Van Sloun Fund for Canine Genetic Research | | Gustavo D Aguirre |
| NIH/NEI | EY-06855 | Gustavo D Aguirre |

The funders had no role in study design, data collection and interpretation, or the decision to submit the work for publication.

## Author contributions

Bilge E Öztürk, Conceptualization, Formal analysis, Investigation, Methodology, Validation, Visualization, Writing – original draft, Writing – review and editing; Molly E Johnson, Conceptualization, Data curation, Formal analysis, Investigation, Methodology, Validation, Visualization, Writing – original draft, Writing – review and editing; Michael Kleyman, Formal analysis, Investigation, Methodology, Visualization, Writing – original draft, Writing – review and editing; Serhan Turunç, Jing He, Sara Jabalameli, Zhouhuan Xi, Meike Visel, Valérie L Dufour, Simone Iwabe, Luis Felipe L Pompeo Marinho, Investigation, Writing – original draft, Writing – review and editing; Gustavo D Aguirre, Funding acquisition, Investigation, Resources, Supervision, Writing – original draft, Writing – review and editing; José-Alain Sahel, Funding acquisition, Supervision, Writing – original draft, Writing – review and editing; David V Schaffer, Conceptualization, Funding acquisition, Resources, Supervision, Writing – original draft, Writing – review and editing; Andreas R Pfenning, Formal analysis, Funding acquisition, Supervision, Writing – original draft, Writing – review and editing; John G Flannery, Funding acquisition, Investigation, Methodology, Supervision, Writing – original draft, Writing – review and editing; William A Beltran, Conceptualization, Funding acquisition, Investigation, Methodology, Supervision, Visualization, Writing – original draft, Writing – review and editing; William R Stauffer, Conceptualization, Formal analysis, Funding acquisition, Investigation, Methodology, Resources, Supervision, Writing – original draft, Writing – review and editing; Leah C Byrne, Conceptualization, Data curation, Formal analysis, Funding acquisition, Investigation, Methodology, Project administration, Resources, Supervision, Validation, Visualization, Writing – original draft, Writing – review and editing

## Author ORCIDs

Bilge E Öztürk http://orcid.org/0000-0001-5117-077X
Molly E Johnson http://orcid.org/0000-0001-9575-3192
Jing He http://orcid.org/0000-0001-9034-8390
Meike Visel http://orcid.org/0000-0002-5033-3730
John G Flannery http://orcid.org/0000-0002-0720-8897
William R Stauffer http://orcid.org/0000-0003-1031-8824
Leah C Byrne http://orcid.org/0000-0002-3229-4993

## Ethics

All procedures were performed in compliance with the ARVO statement for the Use of Animals in Ophthalmic and Vision Research, and for canine studies with approval by the University of Pennsylvania Institutional Animal Care and Use Committee (IACUC # 803813), and for the NHP and mouse studies with approval from the University of Pittsburgh Institutional Animal Care and Use Committee (IACUC #18042326).

## Decision letter and Author response

Decision letter https://doi.org/10.7554/eLife.64175.sa1
Author response https://doi.org/10.7554/eLife.64175.sa2

# Additional files

## Supplementary files

• Supplementary file 1. Table with Summary of injections performed in dogs and primates. AAV selection rounds in canines 2b and 5b were repeated selections of the previous rounds, which did not result in the amplification of AAV variants.

• Supplementary file 2. Statistical Analysis. A. Friedman's test was conducted to determine differences across AAV variants. The test was run separately for each cell type as well as total cells

combined. Marmoset and cynomolgus macaque samples were both used in the analysis (n = 8). Significant *P*-values < 0.05 are shown in bold red. S2.1 p-values resulting from a Friedman's test using percent cells infected as the data points. Significant *P*-values < 0.05 are shown in bold red. S2.2 p-values resulting from a Friedman's test using average transcripts per infected cell as the data points. Significant *P*-values < 0.05 are shown in bold red.

- Supplementary file 3. p-values from a Wilcoxon signed-rank test. A one-sided Wilcoxon signed-rank test was used for a pairwise comparison between K912 or NHP26 and the other variants. P-values were corrected using Benjamini-Hochberg correction method. Significant *P*-values < 0.05 are shown in bold red. S3.1 p-values resulting from a one-sided Wilcoxon signed-rank test using percent cells infected as the data points, comparing NHP12 and other variants. S3.2 p-values resulting from a one-sided Wilcoxon signed-rank test using percent cells infected as the data points, comparing NHP26 and other variants. S3.3 p-values resulting from a one-sided Wilcoxon signed-rank test using average transcripts per infected cell as the data points, comparing K912 and other variants. S3.4 p-values resulting from a one-sided Wilcoxon signed-rank test using average transcripts per infected cell as the data points, comparing NHP26 and other variants.

- Supplementary file 4. List of primers used in the study.

- Transparent reporting form

### Data availability

Data, including count matrix files, raw fastq files as well as AAV/cell barcode tables generated from read quantification, have been uploaded to GEO under accession code GSE161645.

The following dataset was generated:

| Author(s) | Year | Dataset title | Dataset URL | Database and Identifier |
| --- | --- | --- | --- | --- |
| Byrne LC | 2018 | Directed Evolution of AAV for Efficient Gene Delivery to Canine and Primate Retina - Raw counts of variants from deep sequencing | https://doi.org/10.6078/D1895R | Dryad Digital Repository, 10.6078/D1895R |
| Byrne LC | 2021 | scAAVengr, a transcriptome-based pipeline for quantitative ranking of engineered AAVs with single-cell resolution | https://www.ncbi.nlm.nih.gov/geo/query/acc.cgi?acc=GSE161645 | NCBI Gene Expression Omnibus, GSE161645 |

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
