## [Decision Letter]

**Acceptance summary:**

The study entitled "scAAVengr, a transcriptome-based pipeline for quantitative ranking of engineered AAVs with single-cell resolution" by Öztürk et al. describes a method of engineering and identifying adeno-associated viral vectors capable of delivering a transgene to a desired tissue of interest. Using the simian retina as a model, the authors detail their "pipeline" that relies on single cell transcriptome analyses of AAV-transduced cells identified by AAV-encoded green fluorescent protein. The "scAAVengr" technique has the potential to further broaden the available tissue and cell-specific AAV vectors and aid studies developing gene-based therapeutics.

**Decision letter after peer review:**

Thank you for submitting your article "scAAVengr, a transcriptome-based pipeline for quantitative ranking of engineered AAVs with single-cell resolution" for consideration by *eLife*. Your article has been reviewed by 3 peer reviewers, and the evaluation has been overseen by a Reviewing Editor and Huda Zoghbi as the Senior Editor. The reviewers have opted to remain anonymous.

The reviewers have discussed the reviews with one another and the Reviewing Editor has drafted this decision to help you prepare a revised submission.

Summary:

The study entitled "scAAVengr, a transcriptome-based pipeline for quantitative ranking of engineered AAVs with single-cell resolution" by Öztürk et al. describes an approach of engineering and identifying the most efficient AAV capsid/s within a small-scale rAAV library using a barcoded GFP reporter. The approach ("pipeline") relies on a single cell transcriptome analysis of GFP(+), rAAV-transduced cells in simian retina. The analysis assigns a particular internal barcode (capsid's identity) to a particular cell type (external barcodes) constituting retina tissue. The frequency of the respective NGS reads is directly corelated to the capsid transduction efficiency. The approach is indeed a significant technological step forward allowing a reduction in animal use while advancing bioinformatics as a viable and less expensive alternative. The former argument, however, if taken to its extreme, could be counter-productive since, as authors themselves noted (lines 65-66), animals vary, and reducing their number may skew the outcome in favor of a particular animal idiosyncrasies. This paper is of potential interest to a broad audience of investigators studying AAV vectors for therapeutic uses. The scAAVengr technique described is promising, although has not been completely characterized with regard to the number of AAV vectors that can be accurately tested or the use of the approach to study AAV transduction of other organs or tissues beyond the retina.

Essential revisions:

1) Although the technique described is promising, additional characterization is needed to know the number of AAV vectors that can be accurately tested as well as how useful the system is for targeting other tissues beyond retina. For example, the authors suggest that the 17 pooled AAV vectors did not compete with each other to transduce retinal cells, and cite the finding that a small number of sequenced cells were infected with multiple AAVs to support this hypothesis, as shown in Figure 5. But, Figure 5 also shows that the great majority of retinal cells sequenced were infected with a single AAV serotype. Thus, it is not clear that AAVs did not compete. Or does this indicate that most of the "single serotype cells" are the result of a transduction event with a single viral genome? Or that most transduced cells are specifically selective for a specific capsid? Further studies using different numbers of AAV serotypes are needed to address this issue. Similarly, testing scAAVengr in another tissue/organ type would be helpful to demonstrate the potential application of scAAVengr to studies beyond the retina. Alternatively, the paper could be modified to refocus on the use in the retina and data acquired from this focused approach.

(2) One of the major concerns is the description of the pooled, "titer-matched" GFP-barcode libraries (line 506). If, after deep sequencing of the control injected mixture the "relative abundance" of the pool was between ~2.65E+11 – 6.58E+11 vg/ml (~2.5-fold difference, not robust, but within experimental titering error), the resulting "dilution factors" were determined to be in the range of ~14-fold (AAV9 to AAV1). Which of these is correct? If appears the actual NGS read data were normalized by the factor of 14 for AAV1/NHP9 in order to compare them to AAV9. If this was how the data was corrected, is it valid? Once can envision several scenarios (e.g. titer-dependent capsid concentration gradient across the vitreous) for the data misinterpretation.

(3) The manuscript is focused on the scAAVengr method and evaluation of the data described regarding the transduction of retinal cell types with the AAV vectors tested needs to be included in the discussion. For example, why did the K912 vector identified through the DE experiment performed in dogs outperform the NHP vectors in transducing outer retinal cells in the primate experiments reported? Further, what does the finding that the best AAV vectors derived from DE experiments designed to identify vectors that can transduce retinal photoreceptor cells transduced only a small fraction of these cells in the experiments reported? Does it suggest that the barriers to retinal penetration by AAV vectors cannot be overcome by modification of the AAV capsid alone? These are important topics that can inform readers regarding the potential uses of the scAAVengr method and need to be included in the discussion.

(4) There were multiple differences between results obtained in marmoset vs. macaque eyes. For example, more cells were infected in marmoset retinas compared to macaque retina, and fewer "co-infections" with multiple AAVs were detected in macaque retinal cells than marmoset cells. It would be helpful for the authors to discuss the reasons for this, and the related implications for using the scAAVengr method in other organs.

(5) The main combinatorial library, 10-mer amino acid insertion at ~588, is not consistent with several instances describing it as a 7-mer insertion (lines 406, 417, 462 etc). This discrepancy needs addressed.

(6) Normalizing frequencies of the DE-evolving variants to the starting plasmid library (line 107) does not seem to be the best way to follow the enrichment due to the noise from the dead-end non-packaged variants. The original packaged library (Round 0) may be the better common denominator to follow DE enrichment.

(7) The figures used in the manuscript are helpful for presenting the complex data described, but took significant effort to interpret. Inclusion of more complete descriptions of how to "read" the figures in the legends would be helpful.

[Editors' note: further revisions were suggested prior to acceptance, as described below.]

Thank you for submitting your article "scAAVengr, a transcriptome-based pipeline for quantitative ranking of engineered AAVs with single-cell resolution" for consideration by *eLife*. Your article has been reviewed by 1 peer reviewer, and the evaluation has been overseen by a Reviewing Editor and Lu Chen as the Senior Editor. The reviewer has opted to remain anonymous.

Essential revisions:

(1) It is not clear that the newly added experiments in 5B address transduction-competition related to K912. See Reviewer comment 1 for more details.

(2) Ensure consistency in referencing "pools, libraries, mixes" for clarity. See Reviewer comment for more details.

(3) Ensure consistency in referencing "barcodes". See Reviewer comment for more details.

Reviewers comments for the authors:

(1) The newly added experiments described in figure 5B do not really address transduction-competition in a context that is relevant to the main story (the intravitreal injection environment), and does not quantify the transduction events produced by the serotypes relevant to the main story (ie K912).

Example: if K912 was evolved for transduction (TD) of cells in the intravitreal environment, it may no longer be fit to TD HEK293. Therefore quantifying of the TD-efficiency of the would-be competitors is also warranted in order to demonstrate that the model is relevant. Please address this concern by further stating caveats/limitations of this data.

Without testing these variants singly, these experiments do not address whether AAV2's poor performance in the intravitreal environment in the context of the mixed library was due to direct (binding) competition by serotypes that were "more fit", or just AAV2 lack of suitability altogether.

(2) The overall body of work deals with many layers of "mixes, pools, libraries, etc." Phrasing and references to these various entities seemed inconsistent or ambiguous at times leading to significant confusion for this reviewer. I recommend that some space be given to explicitly defining the various groups/mixes/pools, and then maintaining that vocabulary throughout the text. (see line 824, 825. please provide definition of "cloned AAV library" vs "packaged library")

(3) Likewise, this work deals with several layers of "barcoding" (sorted cell barcode, mRNA unique molecular identifier, and viral payload (GFP barcode)). Again, phrasing and references to these various entities seemed inconsistent or ambiguous at times leading to significant confusion for this reviewer. Again, I recommend that some more space be given to definition and graphic representation.

One example of barcode related confusion: It appears that the "GFP barcoding" for variant serotypes do not allow for quantification of multiple infections by the same serotype. (That is for example: all K912 TD events are binned by the same unique barcode associated with the GFP payload, see line 471). This is different from the description of the barcoding of AAV2 in Figure 5C.

---

## [Author Response]

Summary:The study entitled "scAAVengr, a transcriptome-based pipeline for quantitative ranking of engineered AAVs with single-cell resolution" by Öztürk et al. describes an approach of engineering and identifying the most efficient AAV capsid/s within a small-scale rAAV library using a barcoded GFP reporter. The approach ("pipeline") relies on a single cell transcriptome analysis of GFP(+), rAAV-transduced cells in simian retina. The analysis assigns a particular internal barcode (capsid's identity) to a particular cell type (external barcodes) constituting retina tissue. The frequency of the respective NGS reads is directly corelated to the capsid transduction efficiency. The approach is indeed a significant technological step forward allowing a reduction in animal use while advancing bioinformatics as a viable and less expensive alternative. The former argument, however, if taken to its extreme, could be counter-productive since, as authors themselves noted (lines 65-66), animals vary, and reducing their number may skew the outcome in favor of a particular animal idiosyncrasies.

We agree with the reviewers on this point, and we concur that biological replicates will still be necessary to account for individual variability. However, the inclusion of controls in the pooled library quantitatively benchmarks each animal included in the study, allowing for a better understanding of variability and for the identification of outliers.

Essential revisions:(1) Although the technique described is promising, additional characterization is needed to know the number of AAV vectors that can be accurately tested as well as how useful the system is for targeting other tissues beyond retina. For example, the authors suggest that the 17 pooled AAV vectors did not compete with each other to transduce retinal cells, and cite the finding that a small number of sequenced cells were infected with multiple AAVs to support this hypothesis, as shown in Figure 5. But, Figure 5 also shows that the great majority of retinal cells sequenced were infected with a single AAV serotype. Thus, it is not clear that AAVs did not compete. Or does this indicate that most of the "single serotype cells" are the result of a transduction event with a single viral genome? Or that most transduced cells are specifically selective for a specific capsid? Further studies using different numbers of AAV serotypes are needed to address this issue. Similarly, testing scAAVengr in another tissue/organ type would be helpful to demonstrate the potential application of scAAVengr to studies beyond the retina. Alternatively, the paper could be modified to refocus on the use in the retina and data acquired from this focused approach.

We thank the reviewers for these comments. We have addressed these issues with experiments shown in Figure 5B and C and newly created Figures 7 and 8. In order to determine whether the presence of other AAV variants impedes infection of library members, HEK293 cells grown in vitro were infected with either AAV2 alone, or AAV2 in the presence of additional AAV variants (Figure 5B). 1E+6 HEK293 cells were infected with (a) AAV2 (MOI of ~6E+3, 2 technical replicates were performed), or (b) AAV2 (MOI of ~6E+3) + AAV8 (MOI of ~4E+4), AAV9 (MOI ~2E+4) and K912 (MOI ~4E+3), or (c) AAV2 (MOI ~6E+3) and the additional 16 variants of the 17-member library tested in primates (MOI ~5E+3). In all three conditions (alone, in the presence of 3 additional variants, or infected in the presence of 16 additional variants) the number of cells infected by AAV2 and the average number of transcripts recovered from infected cells were stable, indicating that competition for receptors, or the presence of additional variants in the library does not impact quantification of AAV performance.

Then, in order to validate the scAAVengr pipeline in other species and tissues, we screened the same 17-member AAV library in mouse brain, heart and liver following systemic injections (Figure 7). AAV library was packaged, containing each GFP-barcoded virus, and 50 uL of a 5e+12 vg/mL titer library was injected via facial vein in P0 mice. The absolute number of cells infected by each serotype was quantified. Then, the percent of total cells infected by each serotype was quantified for each major cell type and used to create heat maps of serotype infectivity (Figure 8A). Finally, within infected cells, the level of transgene expression was evaluated, relative to total transcripts recovered from each cell (Figure 8B). Each of these metrics was corrected by the dilution factors for variants in the injected library, previously determined by deep sequencing. Heat maps of these metrics revealed that variants AAV8, AAV8-2YF, AAV9, AAV-92YF and AAVrh.10 infected brain, heart and liver following neonatal systemic injections. AAV1 and AAV5 also infected liver cells. These results are in agreement with previously published data on the tropism of these AAV serotypes (Duan, 2016; Foust et al., 2009; L. Wang et al., 2010; Z. Wang et al., 2005; Yang et al., 2014; H. W. Zhang et al., 2011; Zincarelli, Soltys, Rengo, and Rabinowitz, 2008). SCH/NHP26, a variant created through DE in primate with a backbone partially based on AAV9, also infected brain, heart and liver. In contrast, AAV2-based retinal DE variants, including K912 and NHP9, did not efficiently infect organs outside of the eye, indicating that these variants do not cross the vasculature. Together, these experiments demonstrate the application of scAAVengr to studies beyond the retina and in additional animal models. One of the major concerns is the description of the pooled, "titer-matched" GFP-barcode libraries (line 506). If, after deep sequencing of the control injected mixture the "relative abundance" of the pool was between ~2.65E+11 – 6.58E+11 vg/ml (~2.5-fold difference, not robust, but within experimental titering error), the resulting "dilution factors" were determined to be in the range of ~14-fold (AAV9 to AAV1). Which of these is correct? If appears the actual NGS read data were normalized by the factor of 14 for AAV1/NHP9 in order to compare them to AAV9. If this was how the data was corrected, is it valid? Once can envision several scenarios (e.g. titer-dependent capsid concentration gradient across the vitreous) for the data misinterpretation.

We have updated the manuscript to clarify the titers and range of relative abundances of variants in the pool. The manuscript now reads “(The total titer of pooled virus was: ~2E+12 – 5E+12 vg/ml, see Supplementary File 1).” We have removed the dilution factors listed, as these varied from prep to prep. The abundances for all variants in the pool were determined to be within a log of the average variant in the pool. We saw no correlation between abundance in the pool and overall performance (i.e. in some preps the most abundant variant was a low performer and vice versa.) Additionally, we explored the effects of relative abundance of variants in a new experiment, presented in Figure 4B. In this experiment, we saw no significant difference in AAV performance, in terms of percent cells infected or number of cells infected, whether an AAV made up 100% of the library or <10% of the library. The dynamics of virus dilution are an important topic that we are currently exploring further for a manuscript in preparation.

(3) The manuscript is focused on the scAAVengr method and evaluation of the data described regarding the transduction of retinal cell types with the AAV vectors tested needs to be included in the discussion. For example, why did the K912 vector identified through the DE experiment performed in dogs outperform the NHP vectors in transducing outer retinal cells in the primate experiments reported?

We agree that this is an interesting outcome, and we have added the following text to the discussion to address this point**: “**The overall top-performing AAV, K912, was identified through screening performed in the context of the canine retina. Canines, which have an even thicker vitreous than primates, may represent a more difficult model, with a higher barrier for transduction, resulting in higher performing viruses. Further work will need to be done to determine whether vectors from canine screens consistently out-perform variants screened in other animal models.”

Further, what does the finding that the best AAV vectors derived from DE experiments designed to identify vectors that can transduce retinal photoreceptor cells transduced only a small fraction of these cells in the experiments reported? Does it suggest that the barriers to retinal penetration by AAV vectors cannot be overcome by modification of the AAV capsid alone? These are important topics that can inform readers regarding the potential uses of the scAAVengr method and need to be included in the discussion.

We agree that these are important questions, and we have added the following text to the discussion to address these points: “The overall rate of infection achieved by K912, around 2% of total retinal cells, suggests that significant improvements in AAV transduction must be achieved for maximal benefit to be achieved via intravitreal injections. It remains to be seen whether barriers to retinal penetration by AAV vectors from intravitreal injection can be overcome by modification of the AAV capsid alone. The scAAVengr pipeline may also be useful in determining absolute rates of infection and direct comparison with other routes of administration or methods for transgene expression.” There were multiple differences between results obtained in marmoset vs. macaque eyes. For example, more cells were infected in marmoset retinas compared to macaque retina, and fewer "co-infections" with multiple AAVs were detected in macaque retinal cells than marmoset cells. It would be helpful for the authors to discuss the reasons for this, and the related implications for using the scAAVengr method in other organs.

We agree that this is an interesting finding, and we have added the following text to address these points. “Marmoset eyes, which are significantly smaller than human and macaque eyes, were more easily infected than cynomolgus macaques, with greater percentages of cells infected and more infection events per cell. Marmoset eyes may represent a less challenging target for AAV transduction than cynomolgus macaques due to more efficient diffusion of viral particles in smaller eyes, a less rigid vitreous consistency, or immunological factors. This suggests that large primates may be of more use for accurate prediction of performance of gene delivery efficiencies in humans, particularly in the retina, but potentially in other targets as well, such as infection of organs from systemic injections, where diffusion rates and structural barriers may differ between marmosets and macaques.”

The main combinatorial library, 10-mer amino acid insertion at ~588, is not consistent with several instances describing it as a 7-mer insertion (lines 406, 417, 462 etc). This discrepancy needs addressed.

We have updated the text, in order to more accurately describe the library. The following statement was rewritten: “DE was implemented similarly to the screen previously reported in primate retina. AAV2-based libraries, including a ~588 peptide insertion library that contained a random 7-mer peptide flanked by constant flanking linker sequences LA and A, for a total of 10 amino acids.”

Normalizing frequencies of the DE-evolving variants to the starting plasmid library (line 107) does not seem to be the best way to follow the enrichment due to the noise from the dead-end non-packaged variants. The original packaged library (Round 0) may be the better common denominator to follow DE enrichment.

We agree with the reviewers it would be interesting to determine whether Round 0 is a better common denominator to follow DE enrichment, and while this manuscript is focused mainly on the single-cell RNA-seq-based scAAVengr pipeline, we are currently evaluating whether Round 0 may be a better common denominator. This is the focus of a subsequent manuscript in preparation. We have addressed this concern in the manuscript, by including the following text in the discussion: “Here, the best performing variants were determined by using the cloned AAV library as the denominator for evaluation. Additional work will be required to determine whether the packaged library (Round 0) may be a better common denominator to follow DE enrichment.”

7) The figures used in the manuscript are helpful for presenting the complex data described, but took significant effort to interpret. Inclusion of more complete descriptions of how to "read" the figures in the legends would be helpful.

We thank the reviewers for this comment. We have included additional text in figure legends in order to help with interpretation of figures.

[Editors' note: further revisions were suggested prior to acceptance, as described below.]

Reviewers comments for the authors:1) The newly added experiments described in figure 5B do not really address transduction-competition in a context that is relevant to the main story (the intravitreal injection environment), and does not quantify the transduction events produced by the serotypes relevant to the main story (ie K912).Example: if K912 was evolved for transduction (TD) of cells in the intravitreal environment, it may no longer be fit to TD HEK293. Therefore quantifying of the TD-efficiency of the would-be competitors is also warranted in order to demonstrate that the model is relevant. Please address this concern by further stating caveats/limitations of this data.Without testing these variants singly, these experiments do not address whether AAV2's poor performance in the intravitreal environment in the context of the mixed library was due to direct (binding) competition by serotypes that were "more fit", or just AAV2 lack of suitability altogether.

We thank the reviewer for this comment. We agree with this concern, and we have quantified the performance of other variants included in the experiment, including K912. This additional data shows that K912 retains its ability to infect HEK293 cells, and that the performance of K912, as well as the other variants, is constant in this context, despite the presence of additional variants. In addition, we have added the following text to the discussion “However, in vivo confirmation of this finding is warranted, as the barriers affecting the dynamics of infection may differ in a more complex environment, such as the retina.”

2) The overall body of work deals with many layers of "mixes, pools, libraries, etc." Phrasing and references to these various entities seemed inconsistent or ambiguous at times leading to significant confusion for this reviewer. I recommend that some space be given to explicitly defining the various groups/mixes/pools, and then maintaining that vocabulary throughout the text. (see line 824, 825. please provide definition of "cloned AAV library" vs "packaged library")

We thank the reviewer for bringing this to our attention. We have clarified the various libraries used in the manuscript and ensured that our vocabulary is consistent. Additionally, we have added a supplemental figure illustrating the various libraries and pools used throughout the study.

3) Likewise, this work deals with several layers of "barcoding" (sorted cell barcode, mRNA unique molecular identifier, and viral payload (GFP barcode). Again, phrasing and references to these various entities seemed inconsistent or ambiguous at times leading to significant confusion for this reviewer. Again, I recommend that some more space be given to definition and graphic representation.One example of barcode related confusion: It appears that the "GFP barcoding" for variant serotypes do not allow for quantification of multiple infections by the same serotype. (That is for example: all K912 TD events are binned by the same unique barcode associated with the GFP payload, see line 471). This is different from the description of the barcoding of AAV2 in Figure 5C.

It is correct that all K912 TD events are binned by the same barcode. The AAV’s used in the primate scAAVengr study were tracked by AAV-barcodes, with one unique barcode for each AAV serotype. The AAV’s used to infect HEK293 cells in Figure 5C were packaged with random barcodes comprised of n=64, 4,096, 262,144, or 268,435,456 unique members. Thus, the number of unique barcodes recovered indicates the number of different AAVs that infected cells and were recovered through the 10x pipeline. We have clarified the various barcodes used in the manuscript and ensured that our vocabulary is consistent. Additionally, we have added a supplemental figure illustrating the barcoding strategies used throughout the study.